# ABHD5 blunts the sensitivity of colorectal cancer to fluorouracil via promoting autophagic uracil yield

Juanjuan Ou[1], Yuan Peng[1], Weiwen Yang[1], Yue Zhang[1], Jie Hao[1], Fu Li[1], Yanrong Chen[1], Yang Zhao[1], Xiong Xie[1], Shuang Wu[1], Lin Zha[1], Xi Luo[1], Ganfeng Xie[1], Liting Wang[2], Wei Sun[2], Qi Zhou[3], Jianjun Li[1] & Houjie Liang[1]

The efficacy of Fluorouracil (FU) in the treatment of colorectal cancer (CRC) is greatly limited by drug resistance. Autophagy has been implicated in chemoresistance, but the role of selective autophagic degradation in regulating chemoresistance remains unknown. In this study, we revealed a critical role of ABHD5 in charging CRC sensitivity to FU via regulating autophagic uracil yield. We demonstrated that ABHD5 localizes to lysosome and interacts with PDIA5 to prevent PDIA5 from interacting with RNASET2 and inactivating RNASET2. ABHD5 deficiency releases PDIA5 to directly interact with RNASET2 and leave RNASET2 in an inactivate state, which impairs RNASET2-mediated autophagic uracil yield and promotes CRC cells to uptake FU as an exogenous uracil, thus increasing their sensitivity to FU. Our findings for the first time reveal a novel role of ABHD5 in regulating lysosome function, highlighting the significance of ABHD5 as a compelling biomarker predicting the sensitivity of CRCs to FU-based chemotherapy.

[1] Department of Oncology and Southwest Cancer Center, Southwest Hospital, Army Medical University, Chongqing 400038, China. [2] Biomedical Analysis Center, Army Medical University, Chongqing 400038, China. [3] Department of Oncology, Fuling Central Hospital, Chongqing 408099, China. These authors contributed equally: Juanjuan Ou, Yuan Peng, Weiwen Yang, Yue Zhang. Correspondence and requests for materials should be addressed to J.O. (email: ojj521000@sina.com) or to J.L. (email: leejjun2007@163.com) or to H.L. (email: lianghoujie@sina.com)

Colorectal cancer (CRC) has become one of the most common cancers worldwide[1]. Since the early 1990s, fluorouracil (FU), an analogue of uracil, alone or in combination chemotherapy regimes, has been the mainstay chemotherapeutic treatment for CRC patients[2]. FU suppresses pyrimidine synthesis to deplete intracellular dTTP pools by inhibiting thymidylate synthetase, and interferes with nucleoside metabolism to cause cell death via incorporating into RNA and DNA. Although widely used clinically, drug resistance is the main reason greatly limiting the efficacy of FU[3]. Therefore, new strategies for resistance reversal are urgently needed, and understanding the mechanisms by which cancer cells become resistant to FU is an essential step towards predicting or overcoming drug resistance.

Macroautophagy is a catabolic process whereby the intracellular components (e.g., proteins, nucleic acids, and lipids) are degraded by the enzymes in lysosome and recycled[4]. Autophagy has the potential to fuel nearly all aspects of metabolic pathways[5,6], providing cells with tremendous metabolic plasticity. Accumulating findings have shown that autophagy can promote survival under the challenge of chemotherapy, radiotherapy, and targeted agents and thus promotes therapeutic resistance[7–9]. It has been reported that chemotherapy-resistant tumor cells consistently exhibit an enhanced autophagic flux in response to chemotherapy challenge, and manipulation of autophagy would, therefore, be a potential approach to sensitize cancer cells to chemotherapy[10–12], but the key regulatory mechanisms responsible for the increased autophagic flux and autophagic degradation in cancer cells under chemotherapy challenge remains largely unknown.

Bulk degradation via autophagy is principally a non-selective process, however, selective autophagic degradation has recently been realized to play important roles on cell physiology[13]. In rapidly growing cancer cells, the cytoplasmic ribosomes contain almost 50% of all cellular proteins and 80% of total RNA, correlating closely with cell growth rate. Under chemotherapy challenge, ribosome synthesis is immediately stopped and the superfluous ribosomes are degraded. During autophagic degradation of ribosome, not only ribosomal proteins, but also a large amount of ribosomal RNAs are degraded in the autophagolysosome[14–16], but its significance in regulating chemotherapeutic resistance remains unknown.

Metabolic reprogramming and aberrant activity of metabolic enzymes have been characterized as hallmarks of malignant tumors[17]. In our previous study, we have described, a lipolytic factor, ABHD5 (also known as alpha-beta hydrolase domain-containing 5, CGI-58), which functions as an important tumor suppressor in CRCs. We revealed that ABHD5 expression decreases substantially in human CRCs and correlates negatively with malignant features[18]. Importantly, our recent study demonstrated that ABHD5 plays a critical role in maintaining chromosomal stability and protecting genome integrity by regulating autophagy[19]. These findings have been driving us to explore the potential role of ABHD5 in regulating the response of CRCs to chemotherapy.

Here we report that although ABHD5 plays a tumor suppressor role in CRC development and progression, it unexpectedly blunts the sensitivity of CRC cells to FU via promoting RNASET2-mediated autophagic uracil yield. Our findings provide significant insight into the significance of ABHD5 status in predicting the benefit of pMMR patients from FU-based adjuvant chemotherapy.

## Results

### ABHD5 impairs the sensitivity of CRC cells to FU.
To investigate the effect of ABHD5 on the chemotherapeutic response of CRC cells, we first exploited The Genomics of Drug Sensitivity in Cancer Project datasets (GDSC) of CRC cell lines to correlate ABHD5 levels with sensitivity data to chemotherapy-related reagents[20]. Intriguingly, as shown in Fig. 1a, although ABHD5 proficiency only showed a trend toward a positive correlation with IC50 in response to FU in MSI (dMMR) CRC cells, in MSS (pMMR) CRC cells, ABHD5 proficiency exhibited a significant positive correlation with the IC50 to FU. Correspondingly, in the pMMR CRC cell lines SW480 and FET, the IC50 to FU and cell viability under FU challenge were significantly decreased in ABHD5 knockdown cells (Fig. 1b, c, Supplementary Fig. 1a) and substantially increased in ABHD5 overexpression cells (Supplementary Fig. 1b) relative to control cells. In contrast, a minor shift in IC50 value and cell viability in response to oxaliplatin or irinotecan was observed between ABHD5 knockdown and control SW480 cells (Supplementary Fig. 2). Additionally, flow cytometry measurements of Annexin V/7AAD staining (Fig. 1c, d) revealed increased apoptosis in ABHD5 knockdown SW480 cells challenged with FU relative to control cells. To further examine the effect of ABHD5 on the sensitivity of CRC cells to FU, ABHD5 knockdown and control SW480 cells were inoculated intra-abdominally into NOD-SCID mice, and intraperitoneally injected with PBS or FU. Very impressively, compared with the control xenografts, the xenografts derived from ABHD5 knockdown SW480 cells manifested a dramatically increased sensitivity to FU (Fig. 1e), exhibiting significantly increased apoptotic cells in the tumor mass (Fig. 1f). Very intriguingly, in dMMR CRC cells HCT116, manipulation of ABHD5 still showed a relatively modest effect on their sensitivity to FU (Supplementary Fig. 1c and 1d).

Tumor-specific patient-derived xenograft (PDX) models have been shown to retain the intratumoral clonal heterogeneity, chromosomal instability, and histology of the parent tumor through passages in mice. To ascertain the potential clinical relevance of the findings described above, we decided to adapt the intervention trial executed in CRC cell line xenografts to PDX models, which represent a more reliable proxy of prospective findings in patients. We used the selected population of patient-derived pMMR (high expressions of MLH1, MSH2, and MSH6) CRC xenografts (Supplementary Fig. 3), and further divided these xenografts into two subgroups based on their ABHD5 expression proficiency (Supplementary Fig. 3). The mice were randomized into four independent treatment cohorts: (i) pMMR/ABHD5[low] + PBS, (ii) pMMR/ABHD5[high] + PBS, (iii) pMMR/ABHD5[low] + FU, (iv) pMMR/ABHD5[high] + FU. Remarkably, as shown in Fig. 1g, h, pMMR/ABHD5[low] PDX mice benefited significantly from FU treatment, but in contrast, pMMR/ABHD5[high] PDX mice showed resistance to FU and did not benefit from treatment. To further evaluate the effect of treatment with FU, we quantified the presence of proliferative (Ki67) or apoptotic (cleaved caspase 3) cells on histological sections of primary xenograft tumors growing in PDX mice (Fig. 1i). We observed a significant inhibition of proliferation and a significant increase in apoptosis in response to FU in tumors from pMMR/ABHD5[low] PDX; there was no effect on those from pMMR/ABHD5[high] PDX (Fig. 1j).

### ABHD5[high] CRCs do not benefit from FU-based chemotherapy.
CRCs with deficient DNA mismatch repair (dMMR) status have been reported to show improved prognosis but poor responses to FU-based chemotherapy[21]. For the approximately 80% of pMMR patients being conventionally treated with FU-based adjuvant chemotherapy, inconsistencies in chemotherapeutic response are observed even among patients in similar disease stages. The uncertain benefit from the treatment poses a management dilemma[22]. We next queried whether ABHD5

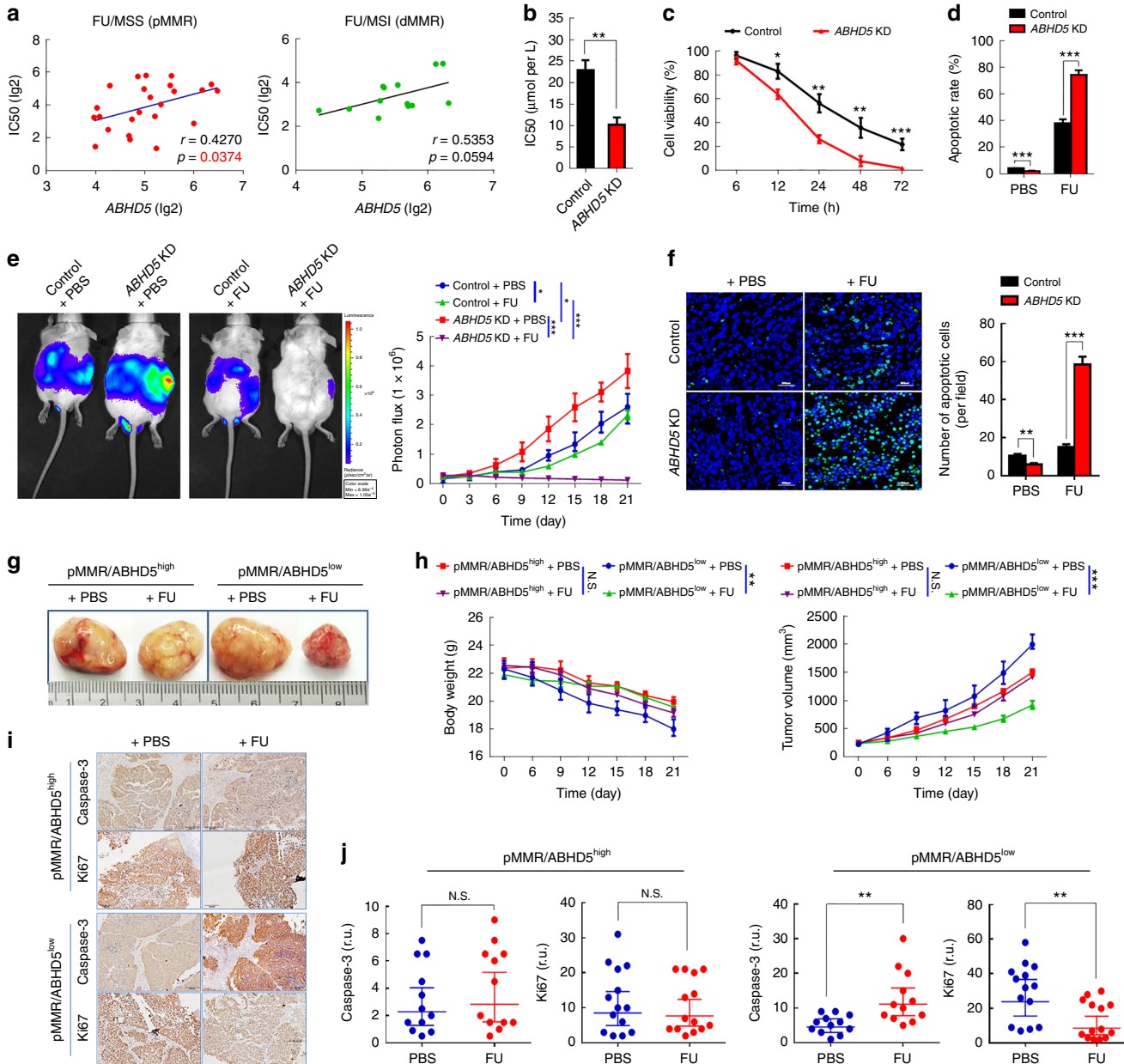

**Fig. 1** ABHD5 impedes the sensitivity of pMMR CRCs to FU. **a** Analyses from the GDSC (Genomics of Drug Sensitivity in Cancer) dataset revealed the correlation between *ABHD5* proficiency and the sensitivity to FU in pMMR or dMMR CRC cell lines (Pearson's correlations). **b** MTT assay determining the IC50 of *ABHD5* knockdown (*ABHD5* KD) and control SW480 cells ($n = 3$, Student's *t*-test). **c** MTT assay determining the cell viability at different time points during FU (25 μM) treatment ($n = 4$, Student's *t*-test). **d** Fluorescence activated cell sorting (FACS) showing the apoptotic rate of cells stained with Annexin V-PE/7AAD following 24 h of treatment with PBS or FU (25 μM) ($n = 4$, Student's *t*-test). **e**, **f** *ABHD5* KD and control SW480 cells were inoculated intra-abdominally into NOD-SCID mice, and intraperitoneal injection (i.p.) of PBS or FU (50 mg per kg) + calcium folinate (80 mg per kg) was administrated once per week for 3 weeks. Tumor burden was measured by bioluminescent imaging ($n = 5$, Two-way ANOVA) (**e**), and the xenografts were stained with TUNEL to calculate apoptotic cells ($n = 5$, Student's *t*-test) (**f**). Scale bar: 200 μm. **g–j** pMMR/ABHD5^high or pMMR/ABHD5^low xenografts were established in PDX mice, and the mice were treated with PBS or FU (50 mg per kg) + calcium folinate (80 mg per kg) (i.p., once per week for 3 weeks) when tumor volume reached about 100 mm³. Representative macroscopic views showing the xenografts in the indicated subgroups (**g**); plot representing the evolution of the body weight and tumor volume of xenografts in the indicated groups ($n = 5$, Two-way ANOVA) (**h**); the proliferating and apoptotic cells in the dissected xenografts were immunostained with Ki67 or caspase-3 (**i**) and analyzed (Horizontal lines indicate arithmetic mean values, and error bars show the 95% CI. r.u., relative units. $n = 12$, Student's *t*-test) (**j**). The quantitative data were presented as mean ± S.D (error bar) (*N.S.* no significant, \**p* < 0.05, \*\**p* < 0.01, \*\*\**p* < 0.001)

might predict the benefit obtained by pMMR CRC patients from FU-based adjuvant chemotherapy. We used the StepMiner algorithm to stratify the population of 361 pMMR CRC patients in the NCBI-GEO dataset into *ABHD5*^high and *ABHD5*^low subgroups, and evaluated the association of *ABHD5* with prognosis

and benefit from FU-based adjuvant chemotherapy in patients with pMMR CRCs. Overall, 99 patients exhibited low *ABHD5* (*ABHD5* = 3.8045–5.3317), and 262 patients exhibited high *ABHD5* (*ABHD5* = 5.3324–7.1488) (Fig. 2a). Patients with *ABHD5*^high tumors were significantly more likely to be *BRAF*

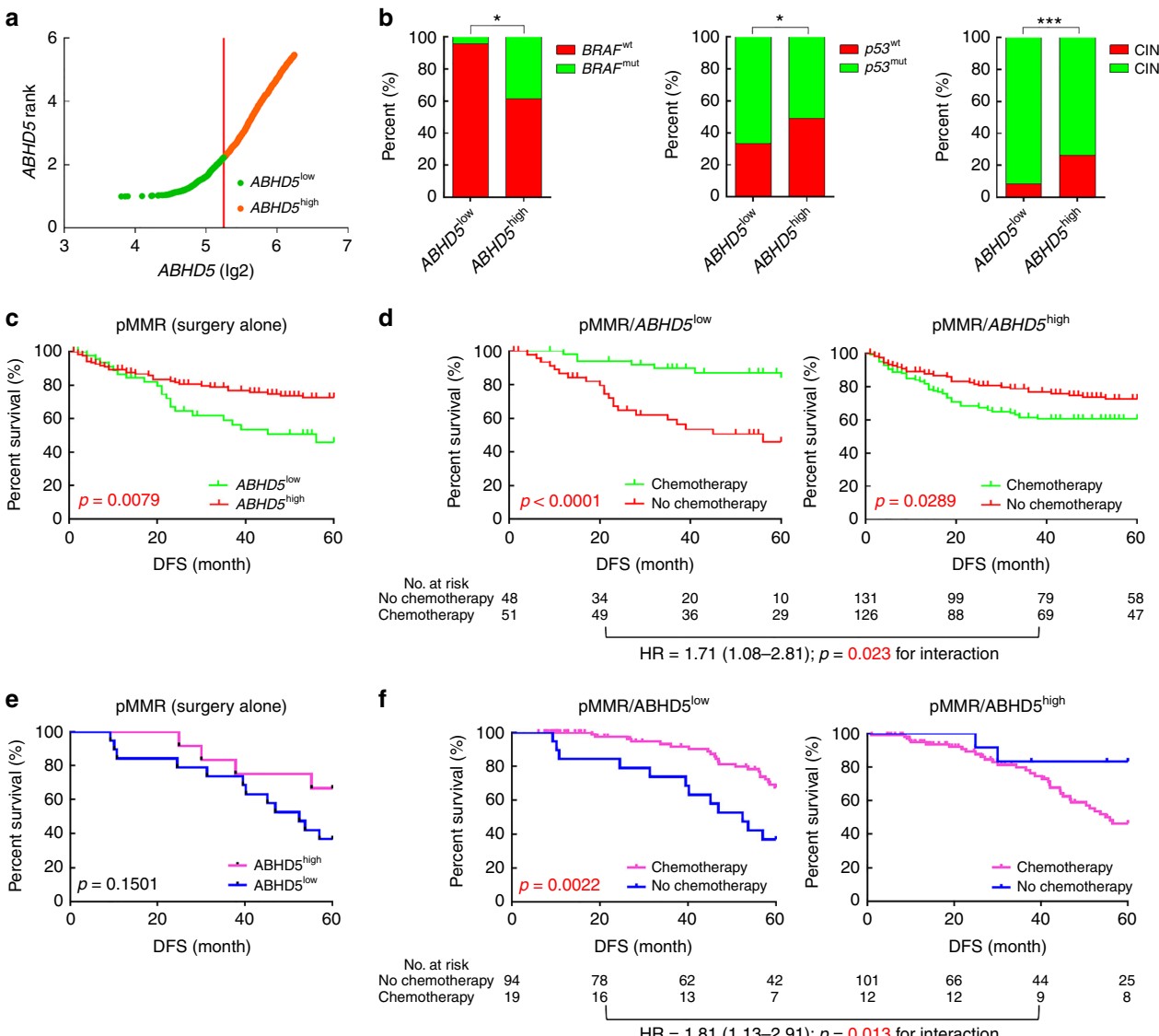

**Fig. 2** ABHD5 expression and the benefit from FU-based adjuvant chemotherapy. **a–d** The StepMiner algorithm was used to stratify the population of 361 pMMR CRC patients (stage II/III) in the NCBI-GEO dataset into *ABHD5*high and *ABHD5*low subgroups, and the association of ABHD5 with prognosis and benefit from FU-based adjuvant chemotherapy was evaluated. **a** Three hundred and sixty-one pMMR CRC patients were stratified into *ABHD5*high and *ABHD5*low subgroups based on mRNA levels (StepMiner). **b** ABHD5high tumors were significantly more likely to be *BRAF* mutant (*BRAF* MT) and *p53* wild type (*p53* WT) and CIN negative (CIN−), and *ABHD5*low tumors were significantly more likely to be *BRAF* wild type (*BRAF* WT) and *p53* mutant (*p53* MT) and CIN positive (CIN+) (*$p < 0.05$, ***$p < 0.001$, $\chi^2$-test). **c** DFS in the subgroup of pMMR/*ABHD5*low or pMMR/*ABHD5*high who received surgery alone (Log-rank test). **d** DFS in pMMR/*ABHD5*low or pMMR/*ABHD5*high subgroup treated with or without FU-based adjuvant chemotherapy (Log-rank test, Cox proportional regression). **e**, **f** A human CRC tissue microarray containing 432 pMMR patient tumor samples from our hospital was analyzed. The samples were stratified into pMMR/ABHD5high or pMMR/ABHD5low subgroups based on the immunostaining score for MMR status and ABHD5 proficiency. **e** DFS in the subgroup of pMMR/ABHD5low or pMMR/ABHD5high who received surgery alone (Log-rank test). **f** DFS in the pMMR/ABHD5low or pMMR/ABHD5high subgroup treated with or without FU-based adjuvant chemotherapy (Log-rank test, Cox proportional regression)

mutation positive and *p53* wild type and chromosomal instability (CIN) negative (Fig. 2b). Among the pMMR patients who received surgery alone, the pMMR/*ABHD5*high subgroup showed a better prognosis compared with the pMMR/*ABHD5*low subgroup (Fig. 2c). Intriguingly, among the pMMR patients who received FU-based adjuvant chemotherapy, the subgroup with *ABHD5*low tumors benefited substantially from adjuvant chemotherapy and achieved a significantly increased probability of DFS relative to those who received surgery alone, while the subgroup with *ABHD5*high tumors did not benefit from adjuvant chemotherapy but instead had a decreased DFS compared with the subgroup of patients who received surgery alone (Fig. 2d).

To further confirm the clinical significance of the above findings, we chose to analyze a human colon cancer tissue microarray collected from the surgery in our hospital. We stratified the patient cohort into two subgroups: pMMR/ABHD5low (176 of 432 patients) and pMMR/ABHD5high (256 of 432 patients). A description of the scoring system and its performance in terms of interobserver agreement is provided in Supplementary Fig. 4. ABHD5low tumors were more enriched in late-stage CRCs, but ABHD5 status showed no correlation with the pathological grade (Supplementary Fig. 5a). Moreover, ABHD5low tumors were associated with a lower rate of survival irrespective of their low or intermediate (G1 or G2) or high (G3) pathological grade - a finding that is consistent with the results of

the multivariate analysis (Supplementary Fig. 5b and 5c). In pMMR patients treated with surgery alone, the pMMR/ABHD5-high subgroup showed a trend toward a prolonged DFS relative to the pMMR/ABHD5low subgroup (Fig. 2e). Expectedly, as shown in Fig. 2f, pMMR/ABHD5low patients benefited significantly from FU-based adjuvant chemotherapy, while pMMR/ABHD5high patients did not benefit.

**ABHD5 impairs FU uptake by promoting autophagic uracil yield.** We next sought to explore the mechanism by which ABHD5 regulates the response of pMMR CRC cells to FU. We first compared the intracellular FU concentrations between ABHD5 knockdown and control SW480 cells. Impressively, the HPLC assay showed that the intracellular level of FU was significantly increased in ABHD5 knockdown SW480 cells relative to control cells (Fig. 3a).

Intracellular drug levels are determined by the drug uptake capacity and drug metabolism efficiency. It has been reported that thymidylate synthase (TS), thymidine phosphorylase (TP) and, dihydropyrimidine dehydrogenase (DPD) control the metabolism of FU. As shown in Fig. 3b, no significant shifts in TS, TP, or DPD were observed between ABHD5 knockdown and control SW480 cells, regardless of FU challenge. We then speculated that the increased intracellular FU level in ABHD5 knockdown cells resulted from an increased drug uptake capacity. Intriguingly, under treatment with FU, the metabolic profile revealed a dramatic decrease in uracil in ABHD5 knockdown SW480 cells relative to control cells (Fig. 3c). These evidence strongly suggest that ABHD5 deficiency may impair the uracil yield and drive pMMR CRC cells to take up FU as an exogenous source of uracil, thus increasing the intracellular FU. Intriguingly, the expression level of carbamoyl phosphate synthetase II (CPS II) and uridine 5'-monophosphate synthase (UMPS), the rate-limiting enzymes responsible for pyrimidine biosynthesis in eukaryotes, showed no shifts between ABHD5 knockdown and control SW480 cells (Fig. 3d), indicating that ABHD5 may affect the uracil yield via a de novo synthesis independent pathway.

To explore the potential mechanism responsible for ABHD5-related uracil yield, we studied 122 human CRCs in GSE38832 dataset. All samples were divided into 2 groups (ABHD5high and ABHD5low) by the median expression value of ABHD5. Impressively, the subtype with low ABHD5 exhibited, on average, decreased levels of lysosome pathway components (Fig. 3e). Moreover, as shown in Fig. 3f, hierarchical clustering analysis based on the expression of ABHD5 identified two clusters characterized by strong and weak expression of the lysosome signature. Based on these evidence, we speculated that autophagic uracil yield mediated by lysosomal RNA degradation may be attributable to the ABHD5-induced increase of uracil in pMMR/ABHD5high CRC cells, thus promoting their resistance to FU. We therefore transiently transfected ABHD5 knockdown and control SW480 cells with the fluorescent-tagged LC3B plasmid[19] to monitor the autophagic flux and kinetics of intracellular FU. Expectedly, a high content screening (HCS) assay showed that the autophagic flux was negatively associated with the intracellular FU levels in a phase-dependent manner (Fig. 3g, h). Remarkably, under FU challenge, Chloroquine (CQ), a potent inhibitor of autophagic flux by targeting autophagosome-lysosome fusion, robustly rescued the intracellular concentrations of uracil and FU (Fig. 3i, j) in ABHD5 overexpression SW480 cells and resensitized the xenograft derived from ABHD5 overexpression SW480 cells to FU (Fig. 3k). These results clearly suggest ABHD5 blunts the response of CRC cells to FU by inducing autophagic uracil yield. Since our previous study has demonstrated that ABHD5

regulates autophagy via activating BECN1[19], we queried whether BECN1 is attributable to ABHD5-induced autophagic uracil yield. Intriguingly, we found that BECN1 activator just modestly reversed the intracellular uracil (Fig. 3l) and the response to FU (Fig. 3m) in ABHD5 knockdown SW480 cells, indicating that ABHD5 regulate autophagic uracil yield via a mechanism beyond BECN1.

**ABHD5 sustains RNASET2 activity to promote autophagic uracil yield.** Since the evidence indicates a critical role of ABHD5 in lysosome function, and it is known that the various hydrolytic enzymes in lysosome is essential for the degradation of macromolecules (e.g., RNA), we deduced that ABHD5 may affect autophagic uracil yield via regulating the activity of ribonucleases (RNases) in lysosome. To determine the potential RNases responsible for ABHD5-induced autophagic uracil yield, we next examined the time-dependent changes in the contents of uracil-related nucleosides and nucleobases under the challenge of FU by ultra-high-performance liquid chromatography-multiple reaction monitoring-mass spectrometry (UHPLC-MRM-MS) analysis (Fig. 4a). Strikingly, contents of cytidine and uridine exhibited remarkable increase in control SW480 cells during FU treatment (Fig. 4b), the intracellular levels increased for up to 3 h. By contrast, little increase was observed in ABHD5 knockdown SW480 cells during FU treatment (Fig. 4b). Correspondingly, the content of uracil showed a remarkable increase in control SW480 cells but a little shift in ABHD5 knockdown SW480 cells under the challenge of FU (Fig. 4b). These evidence suggest that ABHD5 may critically promote the activity of the RNases attributable to the first step of RNA degradation in the lysosome, subsequently resulting in a decrease of autophagic uracil yield.

It was reported that ribonuclease RNASET2 is critically responsible for the first step of ribosomal RNA degradation in autophagolysosome[23,24]. RNASET2 catalyzes the cleavage of RNA through 2',3'-cyclic phosphate intermediates, yielding mono- or oligonucleotides with a terminal 3' phosphate group (Fig. 4c). We thus took note the involvement of RNASET2 in ABHD5-induced autophagic uracil yield during FU treatment. Intriguingly, in ABHD5 knockdown cells, the elevation of 3'-CMP and 3'-UMP, and the consequent nucleosides, was completely abolished, just as exhibited in RNASET2 knockout SW480 cells (Fig. 4d). More impressively, knockdown RNASET2 in ABHD5 overexpression SW480 cells significantly reversed their uracil yield (Fig. 4e) and resensitized them to FU (Fig. 4f, g). These evidence strongly suggest a critical role of RNASET2 in mediating ABHD5-induced autophagic uracil yield.

**ABHD5 protects RNASET2 from being inactivated by PDIA5.** It has been reported that the activity of the RNASET2 and the other lysosomal proteolytic enzyme depends on a low pH[25], we then examined the involvement of lysosomal pH in the effect of ABHD5 on RNASET2 activity. We analyzed the lysosomal pH using an acridine orange (AO) assay. AO is a fluorescent nucleic acid dye that accumulates in acidic spaces such as lysosomes, emitting red light when excited by blue light under the low pH conditions. Relative to the control cells, the red fluorescent signal was found to be dramatically reduced in ABHD5 knockdown cells during the challenge of FU (Supplementary Fig. 6a). The data of the fluorescent intensity ratio demonstrated that the reduction of the red fluorescence signal was due to an increase in lysosomal pH and not a decrease in AO loading (Supplementary Fig. 6b). Cathepsins, including cathepsin B (CSTB) and cathepsin D (CSTD) are lysosomal cysteine proteases and plays an important role in intracellular proteolysis[26,27]. Western blot analysis demonstrated that the loss of ABHD5 downregulated the total

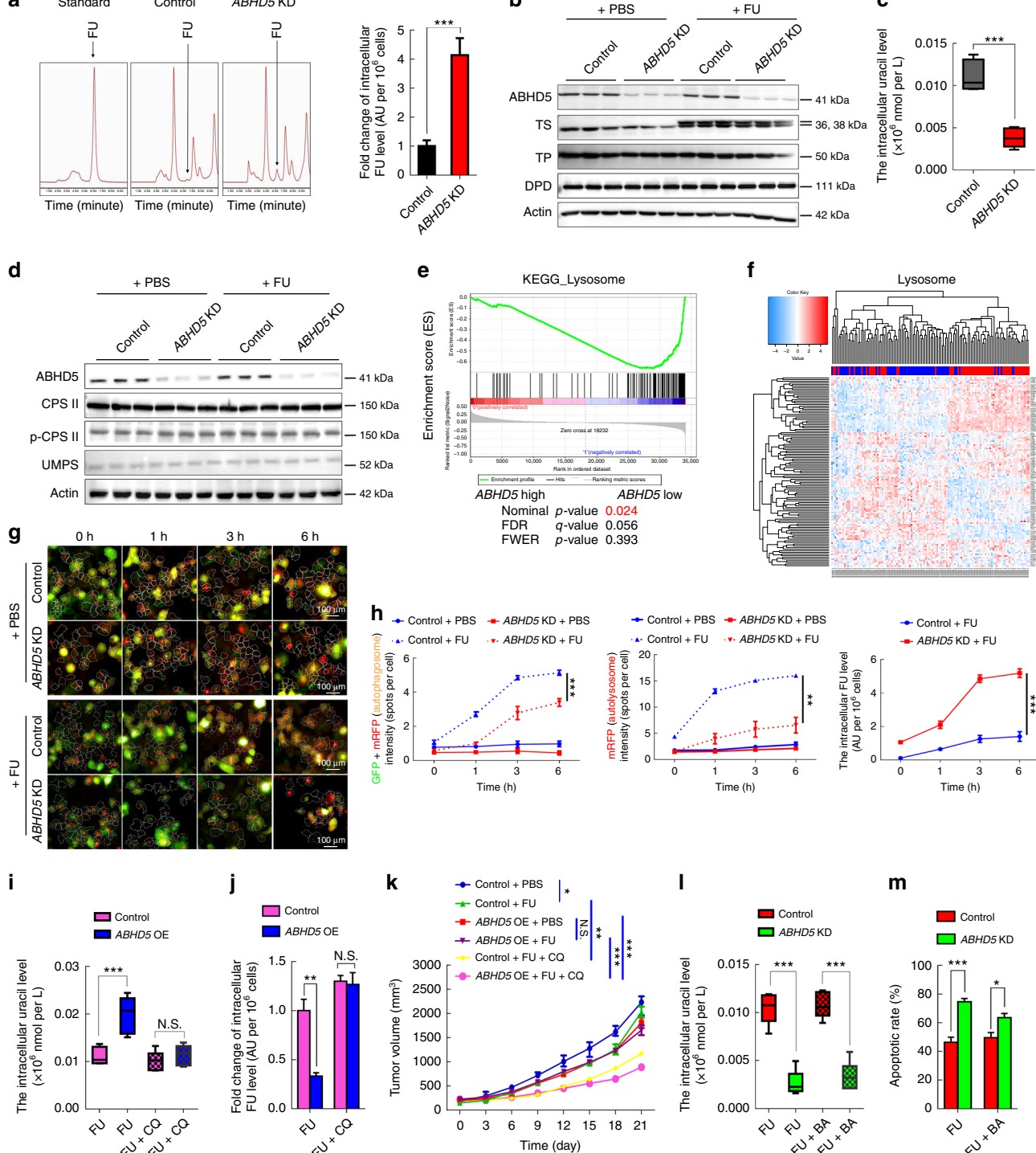

**Fig. 3** ABHD5 proficiency determines the FU uptake of CRC cells. **a** The cells were exposed to PBS or FU (25 μM) for 6 h, and the intracellular concentrations of FU were measured by HPLC ($n = 3$, Student's $t$-test). **b** Western blots of indicated proteins in the cells 24 h following PBS or FU (25 μM) treatment. **c** The cells were exposed to FU (25 μM) for 6 h, and the concentrations of cellular uracil was analyzed by LC/MS ($n = 6$, Student's $t$-test). **d** Western blots of indicated proteins in the cells 24 h after exposure to PBS or FU (25 μM). **e** GSEA analysis of the lysosome pathway. **f** The hierarchical clustering analysis showing the different expression pattern of the genes involved in the lysosome pathway. **g** HCS images showing RFP/GFP labeled LC3 staining in the cells at different time points. **h** Statistical analysis of autophagosome and autolysosome in the cells at different time points, and the corresponding intracellular FU levels are shown ($n = 3$, Two-way ANOVA);. **i, j** The cells were treated with FU (25 μM) alone or FU (25 μM) + CQ (50 μM) for 6 h. The concentrations of cellular uracil (**i**) and the intracellular FU (**j**) were measured ($n = 6$, Student's $t$-test). **k** ABHD5 overexpression (ABHD5 OE) and control cells were inoculated subcutaneously in nude mice, and the mice were treated with PBS, FU (50 mg per kg) + calcium folinate (80 mg per kg) or FU (50 mg per kg) + calcium folinate (80 mg per kg) + CQ (150 mg per kg) (i.p., once per week for 3 weeks). Tumor burden was measured every 3 days ($n = 5$, Two-way ANOVA). **l, m** The cells were treated with FU (25 μM) alone or FU (25 μM) + BECN1 activator (BA) (10 μM) for 6 h (**l**) or 24 h (**m**). The concentrations of intracellular uracil (**l**) and the apoptotic rate (**m**) were analyzed ($n = 6$, Student's $t$-test). The quantitative data were presented as mean ± S.D (error bar) (N.S. no significance, *$p < 0.05$, **$p < 0.01$, ***$p < 0.001$).

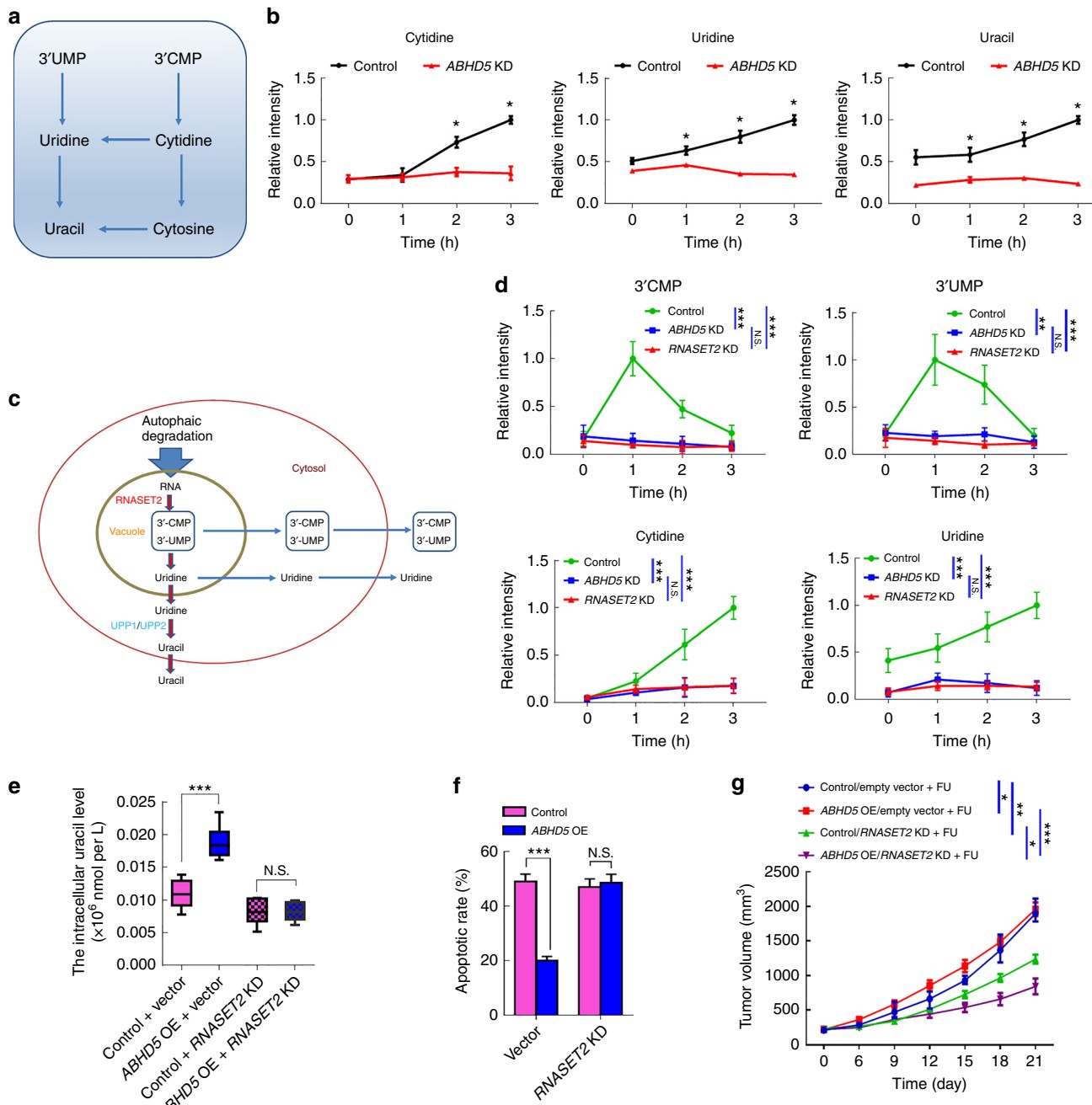

**Fig. 4** RNASET2 mediates ABHD5-induced autophagic uracil yield. **a** Schematic representation of reactions of the uracil-related nucleotide degradation pathway. **b** Nucleosides and nucleobases measurement at the different time points in the cells treated with FU (25 μM). The results are presented as normalized intensities on the basis of the peak height of each metabolite in control cells (*n* = 6, Student's *t*-test). **c** Schematic representation of the pathway for autophagy-dependent RNA degradation in mammalian cells. **d** *RNASET2* knockdown (*RNASET2* KD), *ABHD5* KD, and control SW480 cells were treated with FU (25 μM). Nucleosides and 3′-NMPs were analyzed at different time points. The results are presented as normalized intensities on the basis of the peak height of each metabolite in control cells (*n* = 3, Two-way ANOVA). **e** *RNASET2*-silenced *ABHD5* OE and control SW480 cells were treated with FU (25 μM) for 6 h, and the concentrations of cellular uracil were analyzed (*n* = 6, Student's *t*-test). **f** *RNASET2*-silenced *ABHD5* OE and control SW480 cells were treated with FU (25 μM) for 24 h, and the apoptotic rates were analyzed (*n* = 5, Student's *t*-test). **g** *RNASET2*-silenced *ABHD5* OE and control SW480 were subcutaneously inoculated in nude mice, and the mice were treated with PBS or FU (50 mg per kg) + calcium folinate (80 mg per kg) (i.p., once per week for 3 weeks). Tumor volume was measured every 3 days (*n* = 5, Two-way ANOVA). The quantitative data were presented as mean ± S.D (error bar) (*N.S.* no significance, *\*p* < 0.05, *\*\*p* < 0.01, *\*\*\*p* < 0.001)

protein level of CSTB and CSTD (Supplementary Fig. 6c) as well as their mRNA expression levels (Supplementary Fig. 6d). Meanwhile, the enzymatic activity of both CTSB and CTSD were reduced in *ABHD5* knockdown cells (Supplementary Fig. 6e). It has been reported that ROS production raises the lysosomal pH

and disrupts lysosome activity, and our previous study has demonstrated that ABHD5 deficiency activates the ROS-inflammasome pathway[28]. Intriguingly, although CTSB and CTSD activities in *ABHD5* knockdown cells were dramatically rescued by the anti-ROS agent N-acetyl-cysteine (NAC)

(Supplementary Fig. 6f), very modest effect of NAC on uracil yield was observed (Supplementary Fig. 6g), indicating that lysosomal pH shift is not the predominant mechanism responsible for ABHD5-induced RNASET2 activity.

Very impressively, we found a localization of ABHD5 in lysosome by immunofluorescence staining (Fig. 5a), and ABHD5 expression was also detected in the protein lysis of isolated lysosomes (Fig. 5b), strongly suggesting that ABHD5 may localize

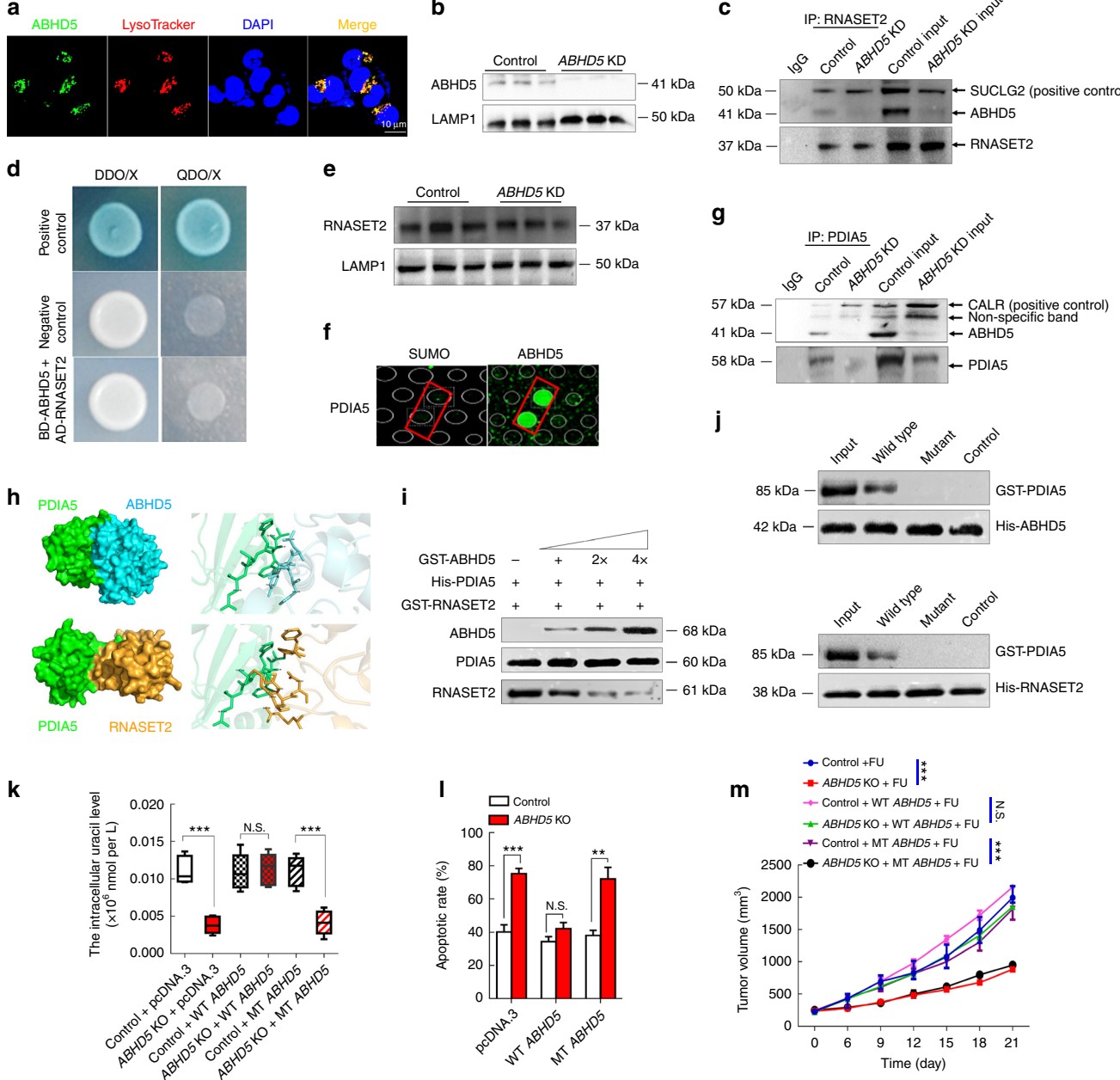

**Fig. 5** ABHD5 sustains the activity of RNASET2. **a** Representative images of immunofluorescent stainings of lysosome (stained by Lysotracker, red) and ABHD5 (green) in SW480 cells. Scale bar: 10 μm. **b** Western blots showing ABHD5 expression in the lysosome lysates of ABHD5 KD and control cells. LAMP1 was used as a reference. **c** Immunoprecipitation showing an interaction between ABHD5 and RNASET2. SUCLG2 was used as a positive control. **d** Yeast two-hybrid assay showing a negative result for the direct interaction between ABHD5 and RNASET2. **e** Western blots of RNASET2 expression in the lysosome lysates of *ABHD5* KD and control cells. **f** A direct interaction between ABHD5 and PDIA5 was shown based on HuProtTM human protein chip. **g** Immunoprecipitation showing an interaction between ABHD5 and PDIA5. CALR was used as a positive control. **h** The interaction complex model between ABHD5, PDIA5, and RNASET2 predicted by protein-protein docking methods. **i** Competitive binding assays of ABHD5 and RNASET2 to PDIA5. **j** In vitro binding assay with wild type PDIA5 (W) or mutant PDIA5 (T) and His-tagged RNASET2 or ABHD5 as indicated. **k** *ABHD5* knockout (ABHD5 KO) and control SW480 cells were transfected with wild type *ABHD5* or mutant *ABHD5* plasmid, and subjected to FU treatment (25 μM) for 6 h. The intracellular uracil was analyzed (n = 6, Student's *t*-test). **l** *ABHD5* KO and control SW480 cells were transfected with wild type *ABHD5* or mutant *ABHD5* plasmid, and subjected to FU treatment (25 μM) for 24 h. The apoptotic rates were analyzed (n = 5, Student's *t*-test). **m** *ABHD5* KO and control SW480 cells transfected with wild type *ABHD5* or mutant *ABHD5* plasmid were subcutaneously inoculated in nude mice, and intraperitoneal injection of PBS or FU (50 mg per kg) + calcium folinate (80 mg per kg) (i.p., once per week for 3 weeks). Tumor volume was measured every 3 days (n = 5, Two-way ANOVA). The quantitative data were presented as mean ± S.D (error bar) (*N.S.* no significance, **p < 0.01, ***p < 0.001)

in lysosome to regulate the activity of RNASET2. Intriguingly, although Co-IP assay detected an interaction between ABHD5 and RNASET2 with a known interactor of RNASET2, Succinate-CoA ligase (SUCLG2)[29], as a positive control (Fig. 5c), the yeast two-hybrid experiments further negated a direct interaction between ABHD5 and RNASET2 (Fig. 5d). In addition, western blots detected no shifts of RNASET2 protein expression levels in lysosome between control and *ABHD5* knockdown SW480 cells (Fig. 5e). These evidence suggest that ABHD5 may not directly regulate the activity of RNASET2. To probe the underlying mechanism, we next used a human ABHD5 recombinant protein to screen its interacting proteins by performing protein-protein interaction experiments based on HuProtTM human protein chip. Remarkably, ABHD5 showed a direct interaction with PDIA5 (Fig. 5f), which is known to inactivate RNASET2 by forming stable disulfide-linked complexes with thermally-unfolded RNASET2[30]. Co-IP assay also confirmed an interaction between ABHD5 and PDIA5 with a known interactor of PDIA5, calreticulin (CALR)[31], as a positive control (Fig. 5g). We then speculated that ABHD5 localizes in lysosome and directly interacts with PDIA5 to prevent PDIA5 from directly interacting with RNASET2 and inactivating RNASET2. In agreement, the interaction pattern between ABHD5, RNASET2 and PDIA5 predicted by protein-protein docking methods[32] showed that a β-sheet structure in PDIA5 was a common domain responsible for the interaction with both ABHD5 and RNASET2 (Fig. 5h). These results indicate that ABHD5 competes with RNASET2 to directly bind to PDIA5. We then analyzed whether ABHD5 affected the binding affinity between PDIA5 and RNASET2 with competitive binding assays of ABHD5 and RNASET2 to PDIA5. Expectedly, as shown in Fig. 5i, the binding affinity of RNASET2 to PDIA5 decreased gradually as ABHD5 increased, indicating that ABHD5 compete with RNASET2 to directly bind to PDIA5, thus affecting the inactivation of RNASET2 by PDIA5. To further determine how ABHD5, PDIA5, and RNASET2 work together, GST-tagged wild type PDIA5 and mutant PDIA5 (ALA-117 ~ GLU-121 deleted) were expressed and purified, and were incubated with His-tagged full-length ABHD5 or RNASET2 on Ni-NTA beads. Immuno-blotting of the Ni-NTA-bound eluates with an anti-GST antibody showed that the domain of PDIA5 (ALA-117 ~ GLU-121) is the common domain responsible for the direct interaction with both ABHD5 and RNASET2 (Fig. 5j). We further constructed a plasmid for the overexpression of mutant *ABHD5*, whose binding region (TYR-193 ~ VAL-197) for the interaction with PDIA5 was deleted. Remarkably, expressing a wild-type *ABHD5* in *ABHD5* knockout SW480 cells strongly reversed their autophagic uracil yield and the sensitivity to FU, whereas transfection of the mutant *ABHD5* failed to do so (Fig. 5k–m). Based on these evidence, we proposed a mechanisms underlying ABHD5-induced FU resistance in CRCs: ABHD5 localizes in lysosome and competes with RNASET2 for directly binding to PDIA5, thus preventing RNASET2 from being inactivated by PDIA5. ABHD5 deficiency leaves RNASET2 in an inactivate state, which impairs RNASET2-induced autophagic uracil yield and promotes CRC cells to uptake FU as an exogenous uracil, thus increasing their sensitivity to FU. In contrast, ABHD5 proficient CRC cells showed an inherent resistance to FU due to an increased autophagic uracil yield (Fig. 6).

**ABHD5 charges autophagic uracil yield independent of PNPLA2.** ABHD5 was well known to be a cofactor of PNPLA2 (patatin like phospholipase domain containing 2). This led us to ask whether ABHD5-mediated autophagic uracil yield and the resistance to FU in CRC cells in a PNPLA2 dependent manner.

We, therefore, exploited GSE59857 dataset to correlate *PNPLA2* expression levels with sensitivity data to chemotherapy-related reagents. As shown in Supplementary Fig. 7a, *PNPLA2* showed no significant correlation with IC50 to FU neither in MSI (dMMR) CRC cells nor in MSS (pMMR) CRC cells. We next stratified the population of 361 pMMR CRC patients in the NCBI-GEO dataset into *PNPLA2*high and *PNPLA2*low subgroups (Supplementary Fig. 7b), and evaluated the association of *PNPLA2* with the benefit from FU-based adjuvant chemotherapy. Among the pMMR patients received surgery alone, the prognosis showed no significant difference between *PNPLA2*high and *PNPLA2*low subgroups (Supplementary Fig. 7c). Among the pMMR patients who received FU-based adjuvant chemotherapy, no significant difference was found between *PNPLA2*high and *PNPLA2*low subgroups (Supplementary Fig. 7c), either.

We further silenced *PNPLA2* in CRC cell line SW480. Intriguingly, relative to control cells, IC50 to FU and the cell viability under FU challenge showed no shifts in *PNPLA2* knockdown cells (Supplementary Fig. 7d). Moreover, the intracellular uracil and the intracellular FU showed no differences between *PNPLA2* knockdown and control cells (Supplementary Fig. 7e). Additionally, no expression of PNPLA2 was detected in the protein lysis from isolated lysosomes (Supplementary Fig. 7f), and Co-IP assay revealed no interaction between PNPLA2 and PDIA5 (Supplementary Fig. 7g). Taken together, these results suggest that ABHD5 promotes RNASET2-mediated autophagic uracil yield independent of PNPLA2.

## Discussion

ABHD5 has been recognized as a co-activator of PNPLA2 in triglyceride degradation for a long time since it was identified from Dorfman syndrome in 1974[33,34]. Our previous study established ABHD5 as a tumor suppressor in CRCs, clarifying an important role of ABHD5 function in the cancer field. In this study, we further revealed a novel and PNPLA2 independent role of ABHD5 in regulating autophagic uracil yield and the sensitivity of CRC cells to the chemotherapeutic agent. Our major discoveries include: (1) discovered previously unappreciated lysosome localization of ABHD5, shedding light on a novel role of ABHD5 in lysosome function; (2) ABHD5 competes with RNASET2 to directly interact with PDIA5; (3) ABHD5 deficiency releases PDIA5 to directly interact with RNASET2 and leave RNASET2 in an inactivate state, which impairs RNASET2-induced autophagic uracil yield; (4) ABHD5 deficiency promotes CRC cells to uptake FU as an exogenous uracil, thus increasing their sensitivity to FU. In contrast, ABHD5 proficient CRC cells showed an inherent resistance to FU due to an increased autophagic uracil yield. We present evidence to show that even in pMMR CRC cells, the ABHD5high subgroup is still relatively resistant to FU, and targeting autophagic uracil yield in pMMR/ABHD5high CRC cells represents a promising new strategy for improving chemotherapeutic efficacy.

Our previous study has shown that loss of ABHD5 in CRCs impairs BECN1-induced autophagic flux and augments genomic instability, which subsequently promotes tumorigenesis[19]. In the present study, we further revealed that ABHD5 localizes in lysosome to promote RNASET2-induced autophagic uracil yield. Hence, we describe a two-step effect of ABHD5 on autophagy, which explains how ABHD5 deficiency promotes CRC tumorigenesis and malignancy while sensitizing CRC cells to FU-based chemotherapy. Of note, although we have demonstrated it is not a direct interaction between ABHD5 and RNASET2, the positive Co-IP result (Fig. 5c) is informative that ABHD5 and RNASET2 are involved in a complex, supporting the localization of ABHD5 to lysosome. The underlying mechanism responsible for the

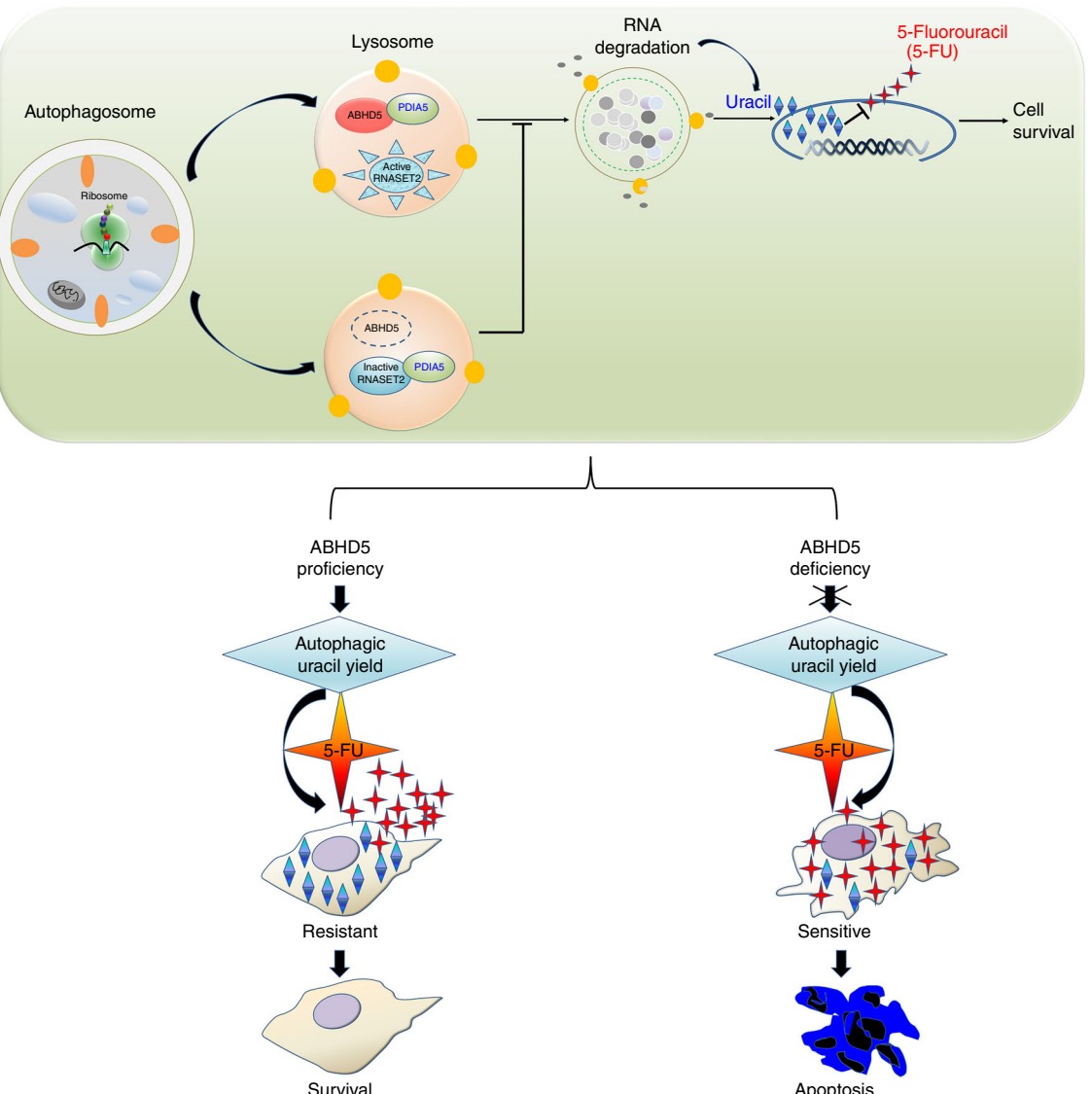

**Fig. 6** The proposed mechanism underlying ABHD5-induced FU resistance. ABHD5 localizes to lysosome and shares share a common interaction domain in PDIA5 with RNASET2. ABHD5 directly interacts with PDIA5 to prevent PDIA5 from directly interacting with RNASET2 and inactivating RNASET2. ABHD5 deficiency releases PDIA5 to directly interact with RNASET2 and leave RNASET2 in an inactivate state, which impairs RNASET2-induced autophagic uracil yield. ABHD5 deficiency promotes CRC cells to uptake FU as an exogenous uracil, thus increasing their sensitivity to FU. In contrast, ABHD5 proficient CRC cells showed an inherent resistance to FU due to an increased autophagic uracil yield

indirect interaction between ABHD5 and RNASET2 could be complicated, warranting further independent investigation.

Autophagy generally serves as a prosurvival mechanism under chemotherapy challenge. As shown in Fig. 4b, the content of uracil showed a remarkable increase in control SW480 cells but a little shift in *ABHD5* knockdown SW480 cells under the challenge of FU. Firstly, it has been reported that FU induces prosurvival autophagy, the increased uracil may be a consequence of the increased autophagic degradation of RNA. Additionally, since FU is an analogue of uracil interfering with nucleoside metabolism via incorporating into RNA and DNA, leading to cytotoxicity and cell death. The increased uracil under FU treatment may due to a compensate mechanism for cell survival, especially in the cancer cells with intact autophagic flux. By contrast, in the cancer cells with impaired autophagic flux, such as ABHD5 deficient cancer cells, FU treatment could not induce neither the prosurvival autophagy nor autophagic degradation of RNA. This can explain why the content of uracil showed a remarkable increase in control

SW480 cells but a little shift in *ABHD5* knockdown SW480 cells under the challenge of FU. To date, different pharmacological autophagy inhibitors have been developed and used in clinical cancer therapy[35–39]. Different autophagy inhibitors block the autophagic process at different stages. For example, Class III PI3K inhibitors (3-methyladenine (3-MA), LY294002 and Wortmannin) are responsible for inhibition of the initiation/expansion stage of autophagy, and antimalarial drugs (CQ or HCQ) or bafilomycin A1 inhibits autophagosome fusion with lysosome and autophagosome degradation during the final stage of autophagy. Thus, different autophagy inhibitors should be selectively subjected to confront different autophagic statuses. Our studies revealed that ABHD5 promotes the autophagic uracil yield by facilitating autophagic RNA degradation, which could explain why CQ significantly improves the efficacy of FU in ABHD5-proficient CRC cells. It is well known that CRC patients with peritoneal metastasis are resistant to chemotherapy with very poor prognosis. Remarkably, as shown in Fig. 1e, FU

efficiently suppressed the growth of *ABHD5* knockdown xenografts in the intra-abdominal xenograft models, indicating that even in the CRC patients with peritoneal metastasis, the combination therapy with FU-based chemotherapy plus CQ may be derived as an efficient strategy to overcome their chemotherapeutic resistance.

It has been demonstrated that autophagy can be regulated by ROS accumulation[40]. Elevated ROS impairs lysosomal maturation and autophagic flux via affecting the lysosomal pH[41]. Our previous study has demonstrated that an ABHD5 deficiency in macrophagy results in an overproduction of ROS[28]. Therefore, we cannot exclude the involvement of ROS-induced impairment of lysosome and autophagic flux under the circumstance of ABHD5 deficiency. In addition, since ROS are important for DNA damage and apoptosis in response to platinum compounds, we can explain why *ABHD5* knockdown can also modestly increase the sensitivity of CRC cells to Oxaliplatin.

Prognostic biomarkers are key to the risk stratification of patients with CRC and the decision to recommend adjuvant chemotherapy in patients with early-stage disease. Our study revealed that pMMR/*ABHD5*low in stage II disease identifies a high-risk subgroup who exhibits a poor prognosis but can benefit from FU-based adjuvant chemotherapy. Moreover, pMMR/*ABHD5*high identifies a low-risk subgroup in stage III disease but provides no benefit from adjuvant chemotherapy. Intriguingly, It is striking that FU-based therapy even promotes tumor progression in ABHD5high CRCs Fig. 2d, f. Actually, chemotherapy is known to show an adverse effect on the immune system of the cancer patients. This can explain why the patients with ABHD5high CRCs, who are resistant to FU-based chemotherapy, did not benefit from FU-based therapy but got a worse outcome due to the toxicity of the chemotherapy to their immune system. Based on this evidence, we propose that stage II CRC patients with pMMR/ABHD5low tumors should be recommended to receive FU-based adjuvant chemotherapy, and stage III CRC patients with pMMR/ABHD5high tumors should be considered to receive FU-based adjuvant chemotherapy combined with a selective autophagy inhibitor (e.g., CQ) to improve the therapeutic efficacy. Given the very limited proportion of the dMMR subpopulation, the stratification based on ABHD5 status is more practical in clinical use. Under chemotherapy challenge, MMR critically contributes to the cell cycle arrest followed by autophagy activation[42,43]. It is therefore not difficult to understand why CRC cells with a dMMR status do not respond efficiently to ABHD5-induced autophagic signaling and to FU. Notably, our findings demonstrated that the manipulation of ABHD5 in dMMR CRC cells still showed a modest effect on their sensitivity to FU. Moreover, as shown in Fig. 1a, even the *p*-value is 0.0594, it is not difficult to find that the correlation between the response to FU and ABHD5 proficiency may be even closer in dMMR cells because the correlation coefficient is 0.53. These evidence indicate that the ABHD5 status, rather than MMR, influences the response of CRCs to chemotherapy. Given the exploratory and retrospective design of our study, our findings require further validation. We advocate for these findings to be confirmed within the framework of randomized clinical trials.

## Methods

**Cells**. Human SW480 and HCT 116 colon cancer cell lines were purchased from the American Type Culture Collection (ATCC) and propagated and passaged as adherent cell cultures according to instructions provided by ATCC. Human FET colon cancer cell line was purchased from BeNa Culture Collection. All cell lines were received as early passages. Cells were maintained in adherent conditions, at 37 °C in humidified atmosphere containing 5% $CO_2$. The medium was changed every other day, cells were passaged using 0.25% trypsin/EDTA (Hyclone) and preserved at early passages. Cell lines were tested mycoplasma contamination in

Cyagen Biosciences Inc. by using their self-developed kit, and were authenticated in Shanghai Biowing Applied Biotechnology Co.Ltd by Sanger sequencing.

**Antibodies and reagents**. Primary antibodies used included anti-ABHD5 (1:1000, Abnova, H00051099-M01), anti-RNASET2 (1:1000, Proteintech, 13753-1-AP), anti-PINPLA2/ATGL (1:1000, Cell Signaling Technology, 2439). anti-thymidine phosphorylase/ECGF1 (1:1000, Cell Signaling Technology, 4307), anti-thymidylate synthase (1:1000, Cell Signaling Technology, 9045), anti-DPD (1:1000, Santa Cruz Biotechnology, sc-376712), anti-CPS-II (1:1000, Santa Cruz Biotechnology, sc-10522), anti-phospho-CPS-II (1:1000, Santa Cruz Biotechnology, sc-377559), anti-UMPS (1:1000, Santa Cruz Biotechnology, sc-103313), anti-Actin (1:5000, Santa Cruz Biotechnology, sc-47778), anti-PDIA5 (1:1000, Santa Cruz Biotechnology, sc-365500), anti-LAMP1 (1:1000, Affinity Biosciences, DF7033), anti-SUCLG2 (1:1000, Novus, nbp1-32521), anti-CTSD (1:500, Boster, PB0019), anti-CTSB (1:500, Boster, BA0428), anti-CALR (1:1000, Abcam, ab92516), anti-MLH1 (1:1000, Abcam, ab92312), anti-MSH2 (1:1000, Abcam, ab70270), anti-MSH6 (1:1000, Abcam, ab92471). The secondary antibodies Alexa Fluor 488 goat anti-mouse IgG (1:5000, Abcam ab150117). DAPI (1:2000, Roche, 10236276001). Chloroquine (C6626) and 5-FU (F6627) were purchased from Sigma. Lysosome Isolation Kit (LYSISO1-1KT), 3'-CMP (CAS 84-52-6) was from Jkchemica, 3'-UMP (CAS 84-53-7) was from BOC Sciences, Uridine (CAS 58-96-8), Cytidine (CAS 65-46-3), Uracil (CAS 66-22-8), Cytosine (CAS 71-30-7) were from Aladdin. Pierce™ Co-Immunoprecipitation Kit (26149) was from Pierce, and Dynabeads™ Protein G Immunoprecipitation Kit (10007D) was from Invitrogen.

**Public datasets**. CRC cell line samples from The Genomics of Drug Sensitivity in Cancer Project (GDSC), and CRC samples collected from Gene Expression Omnibus (GEO) (GSE59857, GSE39582, GSE17538, GSE17536, GSE31595, GSE33113, GSE37892, GSE38832, GSE29623 and GSE39084) were selected according to the following criteria: 1) tumor assayed on Affymetrix Human Genome U133 Plus 2.0 Array; 2) raw data of microarray available; 3) microarray quality control within standards; 4) patients' clinical parameters available. The dataset of The Genomics of Drug Sensitivity in Cancer Project (https://www.cancerrxgene.org/) was used to correlate CRC cell line ABHD5 expression levels with FU sensitivity.

**Gene expression analysis**. The methods used for quality control and raw data processing have been previously described[19]. The dataset was normalized and summarized using robust multi-chip average (RMA) implemented in the R package affy[44], and batch effects were corrected using the ComBat method implemented in the SVA R package[45].

**Calculation of expression thresholds of ABHD5**. We used the StepMiner algorithm to define the gene-array expression thresholds used to separate *ABHD5*low from *ABHD5*high samples. the normalized log2 expression values of *ABHD5* for all samples in the database were ordered from low to high, and a rising step function was fit to the data. The StepMiner algorithm identifies the "step" as the point of the largest jump from low to high values and sets the threshold at the expression value corresponding to the step. An intermediate region is defined around the threshold using a width of 1 (0.5 below and 0.5 above the threshold), corresponding to a 2-fold change in expression, which is the minimum noise level in these large datasets[46,47]. All the samples below the intermediate region (< StepMiner threshold - 0.5) are considered *ABHD5*low, and the rest are considered *ABHD5*high[48].

**Human tissue samples**. Tissue chips consisting of human CRC specimens with chemotherapy and survival follow-up information were collected from the tissue bank in our hospital, and used specifically for analysis of the associations between ABHD5 and survival. All human experiments were approved by the Ethics Committee of Southwest Hospital, Army Medical University.

**MMR status determination**. MMR status of microarray samples was determined by the data resource. Mismatch repair (MMR) tumor status was determined by immunohistochemical analysis (IHC). Tumors with a dMMR phenotype were defined as those showing a loss of expression of one or more MMR proteins by IHC. Proficient MMR phenotype tumors were defined as showing intact MMR protein expression by IHC.

**Immunohistochemistry and immunofluorescence**. For immunohistochenistry, all tissue chip slides were dewaxed and rehydrated. The slides were then incubated in 0.3% $H_2O_2$ in methanol for 30 min to block endogenous peroxidase activity. Antigens were retrieved with 10 mmol/L sodium citrate (pH 6) for 5 min in a pressure cooker. The slides were then incubated with the selected antibody at 4 °C overnight. The slides without treatment of the primary antibody served as negative controls. The slides were developed with an EnVisionTM method (DAKO, Capinteria, CA, USA), visualized using the diaminobenzidine solution, and then lightly counterstained with hematoxylin (H9627, Sigma). All studies involving human subjects were approved by Army Medical University.

For immunofluorescent staining, cell slides were incubated with different primary antibodies, and were subsequently incubated with secondary antibodies. Nuclei were counterstained with DAPI. Images were obtained by confocal laser-scanning microscopy using a LSM780 laser scanning confocal microscope (ZEISS, Germany). To perform image-based analysis for autophagy, cells were infected with the tandem GFP-RFP-LC3 adenovirus for 24 h, and then the cells were treated and imaged for GFP and RFP by using confocal fluorescence microscopy.

**Determination of apoptosis by annexin V/7AAD assay**. Annexin V-APC/7-AAD (KGA1026, KeyGENE bioTECH) was used to detect apoptosis. After drug treatment, cells were treated with 10 μg per mL of annexin V-APC and 1 μg per mL of 7-AAD for 30 min followed by flow cytometry (FACS) using an Accuri™ C6. The strategies account for all FACS sequential gating are graphically provided in Supplementary Figure 8.

**Quantitative RT-PCR**. Total RNA was isolated from cells using the Rneasy Mini Kit (74106, Qiagen). For cDNA synthesis, total RNA was transcribed using PrimeScript (DRR047A, Takara, Dalian, China). The levels of specific RNAs were measured using the ABI 7900 real-time PCR machine and the Fast SybrGreen PCR mastermix according to the manufacturer's instructions. All samples, including the template controls, were assayed in triplicate. The relative number of target transcripts was normalized to GAPDH expression in the same sample. The relative quantification of target gene expression was performed with the standard curve or comparative cycle threshold (CT) method. The primer sequences are listed in the Supplementary Table 1.

**Western blotting**. Proteins were extracted using RIPA Lysis Buffer (P0013, Beyotime, China) and quantified using a BCA kit (P0009, Beyotime, China). Fifty micrograms of each protein sample was separated by 8, 10, or 15% SDS-PAGE and transferred to a polyvinylidene difluoride membrane. The membranes were blocked with 5% BSA and incubated with primary antibodies for 10 h at 4 °C. The membranes were rinsed five times with PBS containing 0.1% Tween 20 and incubated for 1 h with the appropriate horseradish peroxidase-conjugated secondary antibody at 37 °C. Membranes were extensively washed with PBS containing 0.1% Tween 20 three times. The signals were stimulated with enhanced chemiluminescence substrate (NEL105001 EA, PerkinElmer) for 1 min and detected with a Bio-Rad ChemiDoc MP System (170–8280). The primary images (Supplementary Figs. 9-11) were cropped for presentation.

**Pull-down assays**. For Ni-NTA bead pull-down assay, 50 μg of GST, purified wild type GST-PDIA5 or mutant GST-PDIA5 and 50 μg of immobilized His-ABHD5 or His-RNASET2 proteins were added to 500 μl of pulldown buffer (20 mM Tris [pH 8.0], 100 mM NaCl, 0.2% Triton X-100, 1 mM PMSF, 1% protease inhibitor cocktail), then incubated at 4 °C for 4 h. The beads were pelleted and washed three times with 500 μl of 20 mM Tris-HCl (pH 8.0), 200 mM NaCl, 50 mM imidazole and 0.2% Tween 20. The proteins pelleted with the beads were boiled for 5 min at 95 °C and analyzed by SDS-PAGE.

For competitive binding assays, 50 μg of GST-RNASET2 mixed with 0, 50, 100, 200 μg GST-ABHD5 were incubated with 50 μg of immobilized His-PDIA5 at 4 °C for 4 h. Retained proteins were released by adding 2x loading buffer and boiled for 5 min at 95 °C, then resolved by SDS-PAGE and detected by the corresponding antibodies respectively.

**Plasmids, lentiviruses, and recombinant proteins**. Plasmids for overexpression of mutant *ABHD5* and wild type *ABHD5*, lentiviruses for human *RNASET2* knockdown, the recombinant proteins of human wild type His-ABHD5, His-RNASET2, His-PDIA5, GST-PDIA5 and mutant GST-PDIA5 were from Wuhan GeneCreate Biological Engineering Co., Ltd. ABHD5 (Human) recombinant protein (H00051099-P01) and RNASET2 (Human) recombinant protein (H00008635-P01) were purchased from Novus.

**HPLC analysis of intracellular FU**. Concentrations of intracellular FU were analyzed using HPLC analysis. FU was ranged from 0.2 to 100 μmol per L (0.2, 1, 10, 50, and 100 μmol per L) for the preparation of the standard curves. Cells were inoculated in the 6-well plate, and were treated with FU (25 μmol per L) for 6 h. After cell counting, the cells were resuspended in 150 μl PBS and lysed by repeated freeze-thaw at −80 °C and room temperature for 3 times, followed by with a brief sonication in ice-water bath. After incubation at room temperature for 5 min, the samples were centrifuged at 10,000 × g for 15 min. The fresh supernatants were filtered and injected into HPLC (Agilent 7890, France) for analysis. The concentrations of FU was calculated from the respective calibration curves.

**Metabolite extraction**. Cells in logarithmic growth phase were collected onto a membrane filter (0.45 μm, 25 mm; Millipore, Billerica, MA, USA) and washed with 4 ml of 50 mM KCl solution precooled at 4 °C. Cells were rapidly frozen in liquid nitrogen to halt metabolism and stored at −80 °C until extraction. For extraction of cell samples, 1 mL of extraction solvent (methanol:water:chloroform at a 5:2:2 ratio, containing 0.035 μg per mL (+)-10-camphorsulfonic acid, 1 μg per mL PIPES and

0.1 mM methionine sulfone as internal standards) was added to each cell sample and incubated for 30 min at 4 °C. Next, 800 μL of the suspension was transferred to a 1.5 mL centrifuge tube, 400 μL of distilled water was added, and the sample was vortexed. The sample was then centrifuged for 3 min at 4 °C, and 800 μL of the polar extract was collected, filtered (0.2 μm PTFE, Millipore), concentrated to 60 μL, and transferred to glass vials for LC/MS analyses. Medium samples were directly analyzed by LC/MS.

**Measurement of uracil-related nucleosides and nucleobases**. For standard solution preparation, stock solutions were individually prepared by dissolving or diluting each standard substance to give a final concentration of 10 mmol per L. An aliquot of each of the stock solutions was transferred to a 10 mL flask to form a mixed working standard solution. A series of calibration standard solutions were then prepared by stepwise dilution of this mixed standard solution.

After the addition of 400 μL of extraction solution (precooled at −20 °C, acetonitrile-methanol-water, 2:2:1), the samples were vortexed for 30 s, and sonicated for 5 min in ice-water bath. The vortex and sonicate circle was repeated for 3 times, followed by incubation at −20 °C for 1 h and centrifugation at 13,000 × g and 4 °C for 15 min. An 80 μL aliquot of the clear supernatant was transferred to a new EP tube and dried under a gentle nitrogen flow. The residual was reconstituted with 80 μL of water, centrifuged at 13,000 × g and 4 °C for 15 min. The clear supernatant was transferred to an auto-sampler vial for UHPLC-MS/MS analysis.

The UHPLC separation was carried out using an Agilent 1290 Infinity II series UHPLC System (Agilent Technologies), equipped with a Waters ACQUITY UPLC HSS T3 column (100 × 2.1 mm, 1.8 μm). The mobile phase A was 10 mmol per L ammonium acetate/formic acid, and the mobile phase B was methanol. The column temperature was set at 35 °C. The auto-sampler temperature was set at 4 °C and the injection volume was 1 μL. An Agilent 6460 triple quadrupole mass spectrometer (Agilent Technologies), equipped with an AJS electrospray ionization (AJS-ESI) interface, was applied for assay development. Typical ion source parameters were: capillary voltage = +4000/−3500 V, Nozzle Voltage = +500/−500 V, gas (N2) temperature = 300 °C, gas (N2) flow = 5 L per minute, sheath gas (N2) temperature = 250 °C, sheath gas flow = 11 L per minute, nebulizer = 45 psi. The MRM parameters for each of the targeted analytes were optimized using flow injection analysis, by injecting the standard solutions of the individual analytes, into the API source of the mass spectrometer. Agilent MassHunter Work Station Software (B.08.00, Agilent Technologies) was employed for MRM data acquisition and processing. The precision of the quantitation was measured as the relative standard deviation (RSD), determined by injecting analytical replicates of a QC sample. The accuracy of quantitation was measured as the analytical recovery of the QC sample determined. The percent recovery was calculated as (mean observed concentration) * (spiked concentration)$^{-1}$ × 100%.

As to limit of detection (LOD) and limit of quantitation (LOQ), the calibration standard solution was diluted stepwise. These standard solutions were subjected to UHPLC-MRM-MS analysis. The signal-to-noise ratios were used to determine the lower limits of detection (LLODs) and lower limits of quantitation (LLOQs). The LLODs and LLOQs were defined as the analyte concentrations that led to peaks with signal-to-noise ratios of 3 and 10, respectively, according to the US FDA guideline for bioanalytical method validation.

**Cathepsin activity assay**. Cathepsin activity was determined using the commercial Cathepsin D (K143) and Cathepsin B (K140) Activity Fluorometric Assay Kits provided by Biovision according to the manufacturer's protocol. Cathepsin activity was expressed as relative fluorescence units (RFU) per microgram protein.

**In vivo tumor models**. Four-to-six-week-old NOD/SCID male mice and balb/c nude male mice (body weight: 16–20 g) were purchased from the Experimental Animal Center, Institute of Laboratory Animal Sciences (China). The mice were subcutaneously or intra-abdominally injected with control, *ABHD5* overexpression or *ABHD5* knockdown cells (1 × 10⁶ cells per mouse). Intraperitoneal injection of FU-based chemotherapy was administered once per week for 3 weeks. The tumor burden was monitored by bioluminescent imaging or every 3 days, and the mice were sacrificed 3 weeks after treatment. All the animal studies have been approved by the Institutional Animal Care and Use Committee of Army Medical University.

**Establishment of patient-derived tumor xenografts (PDX)**. The 6- to 8-week-old NOD/Shi-scid/IL-2Rγnull (NPI) male mice were maintained under specific pathogen-free conditions and provided with sterile food and water. All the animals were anesthetized with 15 mg per kg of Zoletil® and 2.5 mg per kg of Rompun® by tumor implantation. The human samples were subcutaneously inoculated into the flanks of NPI mice. Following implantation, the mice were monitored once per week. Tumor volumes were calculated using the formula (length * width$^2$) 2$^{-1}$, where length was the longest and width was the shortest axis of the tumor. The mice were euthanized after 3 weeks of FU-based chemotherapy administration following the standard protocol. The xenografts were subsequently transplanted from mouse to mouse or frozen as stocks in nitrogen for further studies. We have complied with all relevant ethical regulations for animal testing and research. All

animal studies were performed with approval from the Animal Care and Use Committee of Army Medical University.

**Statistical analysis**. Continuous variables are reported as the means ± standard deviations and categorical variables as frequencies or percentages. Statistical differences in basal characteristics between groups were analyzed using the $\chi^2$-test for categorical data, applying Yate's correction when required. The quantitative data were analyzed using Student's *t*-test. The influence of different categorical independent variables on the continuous dependent variable was examined by Two-way ANOVA. The correlation between two continuous variables was determined by Pearson correlation coefficient.

Survival curves were generated according to the Kaplan–Meier method, and univariate survival distributions were compared using the log-rank test. Multivariant analysis of hazard risk of death or tumor recurrence was performed using Cox proportional hazards regression. Hazard ratios and 95% confidence intervals (95% CI) for death were computed using Cox survival modeling.

All reported *p*-values are two-sided, and *p*-values less than 0.05 were considered to indicate significance. All calculations were performed using SPSS 13.0 software (SPSS Inc., Chicago, Il, USA).

**Reporting summary**. Further information on experimental design is available in the Nature Research Reporting Summary linked to this article.

## Data availability

The GEO accession numbers reported in this paper are GSE59857, GSE39582, GSE17538, GSE17536, GSE31595, GSE33113, GSE37892, GSE38832, GSE29623 and GSE39084. The other data that support the findings of this study are available within the article and its Supplementary Information files or from the corresponding author upon reasonable request.

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

## Acknowledgements

Grant Support: This work was supported in part by grant numbers 81772647 (J.O.), 81672856 (J.L.), 81672384 (H.L.), 81602628 (X.L.) and 81602423 (L.Z.) from the National Natural Science Foundation of China, cstc2015jcyjB10001 (J.O.) from Basic and Frontier Research Project of Chongqing, cstc2015shmszx120058 (H.L.) from Social Science and Technology Innovation Project of Chongqing and 2008-2-374 (Q.Z.) from The Health Bureau of Chongqing. We got a writing assistance from Nature Research Editing Service, which is supported by grant 81370063 (J.O.). We thank the assistance from Shanghai Biotree Biotech Co., Ltd.

## Author contributions

The authors have made the following declarations about their contributions: Conceived and designed the experiments: J.O., J.L. and H.L. Performed the experiments: Y.P., W.Y., Yue. Z., J.H., F.L., Y.C., Yang. Z., S.W. and Q.Z. Analyzed the data: Y.P., W.Y., L.Z., X.L., S.W., L.W., W.S., X.X., G.X., Q.Z. and J.O. Discussed the data: J.O., J.L. and H.L. Wrote the paper: Y.P., W.Y., J.O., J.L. and H.L.

## Additional information

**Competing interests:** The authors declare no competing interests.

