## [Peer Review File · Nature Communications]

Reviewers' comments:

Reviewer #1 (Remarks to the Author):

In this manuscript, Ou et al. described a new role for lipolytic factor ABHD5 in regulating autophagic uracil yield and the sensitivity of colorectal cancer cells to 5-FU. The major discoveries include: 1) discovered previously unappreciated lysosome localization of ABHD5; 2), ABHD5 competes with RNASET2 for binding to PDIA5, thus preventing RNASET2 from being inactivated by PDIA5; 3), ABHD5 deficiency leaves RNASET2 in an inactivate state, which impairs RNASET2-induced autophagic uracil yield; 4), ABHD5 deficiency promotes CRC cells to uptake FU as an exogenous uracil, thus increasing their sensitivity to 5-FU. The study is very interesting with several novel features and nicely dissected and detailed mechanistic components. While the major findings of this study are potentially significant, I have multiple specific comments that the authors may wish to address before publication.

1. The annotations in the main text does not correspond to the graph in Figure 1 and Figure 3, which is very confusing;
2. The layout of data in Figure 1 would be more logical if Figure 1g and 1j are adjusted to the same pattern in Figure 1h and 1i;
3. Some inconsistent data exist in the Results and Figures:
 - a) The IC₅₀s of control SW480 cells are not consistent in Figure 1b and Figure 6d;
 - b) It is striking that FU therapy even promotes tumor progression in ABHD5-high CRC cells according to Figure 2d, 2f and 4g, while the data are not so consistent in Figure 1h and 3k;
 - c) The stratification of ABHD5 in Figure 2a and that of PNPLA2 in Figure 6b use the same algorithm but the results are not in the same pattern;
 - d) Changes of concentrations of cytidine and uridine over time are not consistent in Figure 4b and 4d;
 - e) In Figure 5i, it is better to describe ABHD5 mutant clones in main text rather than in the figure legend;
4. In Figure 4, it is remarkable that uracil increased in control cells under the challenge of FU, however, no explanation is provided in the Results or Discussion section.
5. There are multiple mistakes or typo errors in data presenting or writing in the manuscript, for example:
 - a) In Figure 1a, the left panel lacks a "log₂" to be parallel with that in the right panel.
 - b) In Figure 1b, "µm/L" should be corrected with "µmol/L" or "µM".
 - c) In line 99, "autophagososomes" should be replaced with "autophagosome".
 - d) In line 109-110, "even these form the therapies themselves" is hard to understand.
 - e) In line 353, "yieldm ediated" should be corrected with "yield mediated".
 - f) In line 390, "attibutable" should be corrected with "attributable".
 - g) In line 721, there misses an "and" between "hundred" and "sixty-one".
 - h) In line 800, there misses a "that" between "cells" and "silenced".
 - i) In line 801, "concentration" should be corrected with "concentrations".

Reviewer #2 (Remarks to the Author):

This manuscript focuses in the very interesting observation that ABHD5 regulates the activity of an enzyme involved in RNA reutilization leading to increase levels of intracellular nucleotides that compete acquisition of exogenous 5-FU thus protecting cancer cell for therapy. The conclusions obtained in this work are relevant in the field of cancer therapy.

However, there are several important issues that need to be addressed in case of publication.

First, the abstract section is a little confusing and should be revised for clarity.

In page 9 it is stated: "very impressively,... in MSS (pMMR) CRC cells, ABHD5 proficiency exhibited a significant positive correlation with the IC50 to FU". This conclusion should be moderated as the correlation, although significant, is not so evident ($r=0.39$). Try to moderate the statements along the manuscript in general, and specially those starting with "impressively", "more impressively" and "very impressively".

Figures from 1g to 1j are not mentioned in the text, which should have been carefully checked before submission. Again, figures 5a-c are incorrectly mentioned in the text as 5a. Authors should understand that these problems difficult a lot the work of the reviewers.

In 5c there is a negative result for RNASET and ABHD5 co-precipitation. However, positive control with a known RNASET interactor is required. In addition, all Co-IP experiments need to be shown in a clearer way including the IPed and the Co-IPed proteins in all the experiments. In the absence of this controls Co-IP experiments are not informative.

Fig 5e, indicated as 5b in the text, is also confusing since in the inputs controls and knocked down (KD) cells look the same. In fact, in none of the figure there is a control showing the extent of the KD and in the cases where there is an image it seems that KD is not working (only in 5b but this is only in the lysosomal fraction). The lack of these controls and the inefficient KD in several experiments is a very important problem that precludes obtaining any further conclusion. Revise all the paper accordingly!

Also the input of the Histidine columns experiment have to be shown.

All Figure 6, including the PNPLA2 data is non-informative and should be included as supplementary or just removed. In figure 6f there is no control for PNPLA2 immunoblot (it is possible that the antibody did not work).

To my view, the model that authors propose is very intriguing but needs further experimental validation.

Dear Reviewers,

Thank you for the careful review of our paper entitled "**ABHD5 Localizes in Lysosome to Sustain RNASET2-Mediated Autophagic Uracil Yield and Blunt the Sensitivity of Colorectal Cancer Cells to 5-Fluorouracil**". We thank you so much for your constructive and thoughtful comments and the opportunity to respond. Those comments are all valuable and very helpful for revising and improving our paper, as well as the important guiding significance to our researches. We have studied the comments carefully and have made correction which we hope meet with approval. The main corrections in the paper and the responds to the reviewer's comments are enumerated below:

Reviewer 1:

Comments to the authors:

In this manuscript, Ou et al. described a new role for lipolytic factor ABHD5 in regulating autophagic uracil yield and the sensitivity of colorectal cancer cells to 5-FU. The major discoveries include: 1) discovered previously unappreciated lysosome localization of ABHD5; 2) ABHD5 competes with RNASET2 for binding to PDIA5, thus preventing RNASET2 from being inactivated by PDIA5; 3) ABHD5 deficiency leaves RNASET2 in an inactivate state, which impairs RNASET2-induced autophagic uracil yield; 4) ABHD5 deficiency promotes CRC cells to uptake 5-FU as an exogenous uracil, thus increasing their sensitivity to 5-FU. The study is very interesting with several

novel features and nicely dissected and detailed mechanistic components. While the major findings of this study are potentially significant, I have multiple specific comments that the authors may wish to address before publication.

1. The annotations in the main text does not correspond to the graph in Figure 1 and Figure 3, which is very confusing;

Response-- We are sorry for these errors, and have revised the annotations in the main text to match the corresponding graph.

2. The layout of data in Figure 1 would be more logical if Figure 1g and 1j are adjusted to the same pattern in Figure 1h and 1i;

Response-- We have adjusted the layout of data in Fig. 1g and 1j to the same pattern in Fig. 1h and 1i.

3. Some inconsistent data exist in the Results and Figures:

a) The IC50s of control SW480 cells are not consistent in Figure 1b and Figure 6d;

Response—We thank the reviewer for raising this question. Since we used different vectors for ABHD5 knockdown and PNPLA2 knockdown in our precious study, we can not exclude the effect of the different vectors on the IC50 of SW480 cells. To avoid this problem, we reconstructed the same vector system for PNPLA2 knockdown and ABHD5 knockdown, and reperformed these experiments. To unify the data, we used the MTT assay for both IC50s and cell viability. Expectedly, we got the consistent values by using the same vector system, and have revised the corresponding data.

b) It is striking that FU therapy even promotes tumor progression in ABHD5-high CRC cells according to Figure 2d, 2f and 4g, while the data are not so consistent in Figure 1h and 3k;

Response—We appreciate the reviewer for raising this good question. We know that chemotherapy shows an adverse effect on the immune system of the cancer patients. This can explain why in Fig. 2d and 2f, the patients with ABHD5-high CRCs, who are resistant to 5-FU-based chemotherapy, did not benefit from 5-FU-based therapy but got a worse outcome due to the toxicity of the chemotherapy to their immune system. In Fig. 1h and 3k, the PDX models we used are deficient of an intact immune system, which can not mimic the damage of chemotherapy to the immune system. This can explain why the adverse effect of 5-FU-based therapy to the immune system and the prognosis could not be reflected as observed in the clinical patients. We have discussed this in the revised manuscript. To avoid confusing the readers, we removed the survival data from PDX mouse model in the revised version, leaving the data monitoring body weight and tumor volume, which is still sufficient to reflect the sensitivity of pMMR CRCs with varied ABHD5 proficiency to 5-FU-based chemotherapy. Regarding the previous data presented in Fig. 4g, the growth curves of control and ABHD5 knockdown xenografts without 5-FU treatment are not shown. Actually, ABHD5 overexpression CRC cells showed a relatively decreased growth relative to the control cells. On the other hand, ABHD5 overexpression CRC cells are highly resistant to 5-FU while the control cells

are still relatively sensitive. This can explain why the tumor growth of ABHD5 overexpression xenografts is modestly faster than the control xenografts under 5-FU challenge.

In addition, as shown in Fig. 3k in the revised manuscript, to better demonstrate the role of chloroquine in reversing the resistance of ABHD5-overexpression CRC cells to 5-FU, we adjusted the dosage of chloroquine (from 80mg/kg to 150mg/kg) and reperformed the experiments on the subcutaneous xenograft model in balb/c nude mice. Meanwhile, in the revised Fig. 4g, we reselected all the cells by antibiotic selection and reperformed the experiments to better demonstrate the effect of RNASET2 silencing in reversing the resistance of ABHD5 overexpression CRC cells to 5-FU *in vivo*.

c) The stratification of ABHD5 in Figure 2a and that of PNPLA2 in Figure 6b use the same algorithm but the results are not in the same pattern;

Response-- We have adjusted the stratification of ABHD5 in Fig. 2a and that of PNPLA2 in Fig. 6b to the same pattern.

d) Changes of concentrations of cytidine and uridine over time are not consistent in Figure 4b and 4d;

Response-- We checked our raw data, and found that the data in Fig. 4d have been shown up in a disordered pattern due to an error occurred while plotting the figures by using graphpad software. To confirm these results, we reperformed LC/MS assays, and present the revised data without changing

our conclusion.

e) In Figure 5i, it is better to describe ABHD5 mutant clones in main text rather than in the figure legend;

Response-- We have revised this following the reviewer's suggestion.

4. In Figure 4, it is remarkable that uracil increased in control cells under the challenge of FU, however, no explanation is provided in the Results or Discussion section.

Response—We appreciate the reviewer for raising this good question. As shown in Fig. 4b, the content of uracil showed a remarkable increase in control SW480 cells but a little shift in ABHD5 knockdown SW480 cells under the challenge of 5-FU. Firstly, it has been reported that 5-FU induces prosurvival autophagy, the increased uracil may be a consequence from the increased autophagic degradation of RNA. Additionally, since 5-FU is an analogue of uracil interfering with nucleoside metabolism via incorporating into RNA and DNA, leading to cytotoxicity and cell death. The increased uracil under 5-FU treatment may due to a compensate mechanism for cell survival, especially in the cancer cells with intact autophagic flux. By contrast, in the cancer cells with impaired autophagic flux, such as ABHD5 deficient cancer cells, 5-FU treatment could not induce neither the prosurvival autophagy nor autophagic degradation of RNA. This can explain why the content of uracil showed a remarkable increase in control SW480 cells but a little shift in ABHD5 knockdown SW480 cells under the challenge of 5-FU. We have added this

explanation in Discussion section in the revised manuscript.

5. There are multiple mistakes or typo errors in data presenting or writing in the manuscript, for example:

a) In Figure 1a, the left panel lacks a “log2” to be parallel with that in the right panel.

b) In Figure 1b, “ $\mu\text{m/L}$ ” should be corrected with “ $\mu\text{mol/L}$ ” or “ μM ”.

c) In line 99, “autophagososomes” should be replaced with “autophagosome”.

d) In line 109-110, “even these form the therapies themselves” is hard to understand.

e) In line 353, “yieldm ediated” should be corrected with “yield mediated”.

f) In line 390, “attibutable” should be corrected with “attributable”.

g) In line 721, there misses an “and” between “hundred” and “sixty-one”.

h) In line 800, there misses a “that” between “cells” and “silenced”.

i) In line 801, “concentration” should be corrected with “concentrations”.

Response-- We are sorry for these errors and have revised them along the manuscript.

Reviewer #2 (Remarks to the Author):

This manuscript focuses in the very interesting observation that ABHD5 regulates the activity of an enzyme involved in RNA reutilization leading to increase levels of intracellular nucleotides that compete acquisition of exogenous 5-FU thus protecting cancer cell for therapy. The conclusions

obtained in this work are relevant in the field of cancer therapy. However, there are several important issues that need to be addressed in case of publication.

First, the abstract section is a little confusing and should be revised for clarity.

Response-- We appreciate the reviewer's suggestion, and we have revised the abstract section.

In page 9 it is stated: "very impressively,... in MSS (pMMR) CRC cells, ABHD5 proficiency exhibited a significant positive correlation with the IC50 to FU". This conclusion should be moderated as the correlation, although significant, is not so evident ($r=0.39$). Try to moderate the statements along the manuscript in general, and specially those starting with "impressively", "more impressively" and "very impressively".

Response-- We fully accept the reviewer's suggestion, and we have revised these statements.

Figures from 1g to 1j are not mentioned in the text, which should have been carefully checked before submission. Again, figures 5a-c are incorrectly mentioned in the text as 5a. Authors should understand that these problems difficult a lot the work of the reviewers.

Response-- We are very sorry for these errors, and have revised the annotations in the main text to match the corresponding graph.

In 5c there is a negative result for RNASET and ABHD5 co-precipitation. However, positive control with a known RNASET interactor is required. In addition, all Co-IP experiments need to be shown in a clearer way including the

IPed and the Co-IPed proteins in all the experiments. In the absence of this controls Co-IP experiments are not informative.

Response-- We do appreciate the reviewer's good suggestion and instruction. We reformed these Co-IP experiments as requested. Unexpectedly, as shown in Fig. 5c, we found a positive result for RNASET2 and ABHD5 co-precipitation this time with a known interactor of RNASET2, SUCLG2, as a positive control. To make sure the authenticity of the results, we repeated this Co-IP for 3 times, and got the consistent results. We compared all the reagents we used between the previous experiments and the present experiments, and found that only the co-immunoprecipitation kits are different. The kit for the previous experiments is Pierce™ Co-Immunoprecipitation Kit (cat # 26149, Pierce), and the kit for the present experiments is Dynabeads™ Protein G Immunoprecipitation Kit (cat # 10007D, Invitrogen). We next used Yeast Two-Hybrid experiments to determine whether a direct interaction exists between ABHD5 and RNASET2, but got a negative result (Fig. 5d). In addition, western blots detected no shift of RNASET2 expression levels in lysosome lysates between control and ABHD5 knockdown cells. These evidence strongly suggest that ABHD5 may not directly regulates the activity of RNASET2. To probe the potential mediator between ABHD5 and RNASET2, we next used a human ABHD5 recombinant protein to screen its interacting proteins by performing protein-protein interaction experiments based on HuProt™ human protein chip. Remarkably, ABHD5 showed an interaction

with PDIA5 (Fig. 5f), which is known to inactivate RNASET2 by forming stable disulfide-linked complexes with thermally-unfolded RNASET2. We have revised this part in the manuscript accordingly.

Fig 5e, indicated as 5b in the text, is also confusing since in the inputs controls and knocked down (KD) cells look the same. In fact, in none of the figure there is a control showing the extent of the KD and in the cases where there is an image it seems that KD is not working (only in 5b but this is only in the lysosomal fraction). The lack of these controls and the inefficient KD in several experiments is a very important problem that precludes obtaining any further conclusion. Revise all the paper accordingly!

Response-- We fully accept and appreciate the reviewer's good suggestion and instruction. We reformed these Co-IP experiments as requested by using the cells fully selected by puromycin. As shown in Fig. 5c and Fig. 5g in the revised version, an efficient KD of ABHD5 is presented accompanied with the expected shifts of CO-IP between ABHD5, PDIA5 and RNASET2.

Also the input of the Histidine columns experiment have to be shown.

Response-- We reformed the Histidine columns experiment following the reviewer's suggestion, which is shown in Fig. 5j in the revised version.

All Figure 6, including the PNPLA2 data is non-informative and should be included as supplementary or just removed. In figure 6f there is no control for PNPLA2 immunoblot (it is possible that the antibody did not work).

Response-- We thank the reviewer's good suggestion. In the revised

manuscript, the PNPLA2 data is included as Supplementary Fig. 7. In Supplementary Fig. 7f, the expression of PNPLA2 can be detected in whole lysates of control and PNPLA2 knockdown cells but not in lysosome lysates, indicating that PNPLA2 can not localize in lysosome as ABHD5 does.

To my view, the model that authors propose is very intriguing but needs further experimental validation.

Reviewers' comments:

Reviewer #1 (Remarks to the Author):

The authors have sufficiently addressed all my concerns. I have no more questions.

Reviewer #2 (Remarks to the Author):

The manuscript is interesting and has been improved according to reviewers' comments. Said this, there are several issues that need to be further revised:

ABHD5 western blot panels are still absent from figures 3B and 3D where the extent of the knock down is essential for interpretation of the results.

In Figure 5C it is shown that ARNAsen interacts with ABHD5, which is not in accordance with the model proposed. Authors need to explain in the text their interpretation of this result?

As a general rule, the proteins that are primary precipitated are shown in addition to the co-precipitates to ensure comparable levels of precipitation in all conditions. This is lacking in figures 5c, 5d and 5i. This was already requested in the first round of review.

Also, authors show controls of co-precipitation in the same blot as the experimental proteins (5c and 5g). Have they incubated the membranes with both antibodies together?

In the in vitro interactions shown in figure 5j the molecular weight of the proteins in the input fraction and the precipitates is totally divergent, which is at least surprising. This result needs to be mentioned and justified.

Dear Reviewers,

Thank you for the careful review of our paper entitled “**ABHD5 Localizes in Lysosome to Sustain RNASET2-Mediated Autophagic Uracil Yield and Blunt the Sensitivity of Colorectal Cancer Cells to 5-Fluorouracil**”. We thank you so much for your constructive and thoughtful comments and the opportunity to respond. Your insightful comments are not only valuable and helpful for improving this manuscript but also of guiding significance to our future study. Now, following your suggestions, we have further revised the manuscript carefully. The main corrections in the paper and the responds to the reviewer’s comments are enumerated below:

Reviewer #1(Remarks to the Author):

The authors have sufficiently addressed all my concerns. I have no more questions.

Reviewer #2 (Remarks to the Author):

The manuscript is interesting and has been improved according to reviewers’ comments. Said this, there are several issues that need to be further revised:

Question 1: ABHD5 western blot panels are still absent from figures 3B and 3D where the extent of the knock down is essential for interpretation of the results.

Response-- We apologize for missing these two results during the revision. We have reperformed the western blots in fig. 3b and 3d containing the ABHD5 western blot panels as requested.

Question2: In Figure 5C it is shown that RNAsset interacts with ABHD5, which is not in accordance with the model proposed. Authors need to explain in the text their interpretation of this result?

Response-- We do appreciate and fully accept the reviewer’s suggestion and have discussed further on this point in the revised manuscript. As we have

explained in the first round revision, the false-negative result of Co-IP for ABHD5 and RNASET2 was most likely due to the different co-immunoprecipitation kits we used. The kit for the previous experiments is Pierce™ Co-Immunoprecipitation Kit (cat # 26149, Pierce), and the kit for the present experiments is Dynabeads™ Protein G Immunoprecipitation Kit (cat # 10007D, Invitrogen). Although our newly performed Co-IP assays detected an interaction between ABHD5 and RNASET2, the subsequent yeast two-hybrid experiments further negated a direct interaction between ABHD5 and RNASET2. These results demonstrated that even ABHD5 and RNASET2 may be involved in a complex, they are not directly interacted. The underlying mechanism responsible for the indirect interaction between ABHD5 and RNASET2 could be complicated, warranting further independent investigation.

To exactly confirm the proposed model that ABHD5 competes with RNASET2 to directly interact with PDIA5, we have replaced the previous data in Fig. 5i with competitive binding assays of ABHD5 and RNASET2 to PDIA5. As shown in revised Fig. 5i, the binding affinity of RNASET2 to PDIA5 decreased gradually as ABHD5 increased, indicating that ABHD5 compete with RNASET2 to bind directly to PDIA5, thus affecting the inactivation of RNASET2 by PDIA5.

These evidence still support the model we previously proposed: ABHD5 competes with RNASET2 to directly interact with PDIA5 in lysosome. ABHD5 deficiency releases PDIA5 to directly interact with RNASET2 and leave RNASET2 in an inactivate state, which impairs RNASET2-induced autophagic uracil yield and promotes CRC cells to uptake 5-FU as an exogenous uracil, thus increasing their sensitivity to 5-FU. In contrast, in ABHD5 proficient CRC cells, even PDIA5 and RNASET2 could be involved in a complex via ABHD5, RNASET2 could not be directly reached and inactivated by PDIA5 because the domain in PDIA5 responsible for the direct interaction with RNASET2 is occupied by ABHD5, and the RNASET2-induced autophagic uracil yield in

ABHD5 proficient CRC cells is therefore intact and results in a resistance to 5-FU uptake.

We appreciate the reviewer for reminding us in this regard. Accordingly, we have revised the corresponding description in the revised manuscript to clarify our model.

Question3: As a general rule, the proteins that are primary precipitated are shown in addition to the co-precipitates to ensure comparable levels of precipitation in all conditions. This is lacking in figures 5c, 5d and 5i. This was already requested in the first round of review.

Response-- We are sorry for our negligence. Actually, we have already immunoblotted both the precipitated and co-precipitated proteins as requested in the first round of revision but forgot to present the primary precipitated proteins because we were focusing on showing the positive controls. We have now added the blots of the primary precipitated proteins into fig. 5c and 5g in the revised manuscript. Of additional note, we have replaced the previous data in Fig. 5i with the competitive binding assays of ABHD5 and RNASET2 to PDIA5.

Question 4: Also, authors show controls of co-precipitation in the same blot as the experimental proteins (5c and 5g). Have they incubated the membranes with both antibodies together?

Response-- We thank the reviewer for this question. Since the indicated proteins can be well separated in the same blot based on their molecular weight, we incubated the membranes with both antibodies together.

Question5: The difference in molecular weight between the two proteins is relatively large. In the in vitro interactions shown in figure 5j the molecular weight of the proteins in the input fraction and the precipitates is totally

divergent, which is at least surprising. This result need to be mentioned and justified.

Response-- We appreciate the reviewer for raising this question. We previously used the non-denaturing conditions to elute and run the recombinant proteins in order to retrieve the proteins. Nevertheless, we speculate that, in the non-denaturing conditions, the interaction between the two proteins may not be interrupted. As a consequence, the molecular weight of the pull-down protein might be larger than the input protein due to the interaction between the two proteins.

Anticipating that the non-denaturing conditions might not be suitable for pull-down assay because the detected bands could be divergent, we therefore changed our experiment conditions to exactly follow the standard pull-down protocol. As expected, we got the precipitates with the similar molecular weight to the input fraction. Of additional note, while we previously produced these recombinant proteins with different tags for different pull-down assays, we used the GST-tagged PDIA5 recombinant protein this time because the Flag-tagged PDIA5 was found to be severely degraded.

REVIEWERS' COMMENTS:

Reviewer #2 (Remarks to the Author):

Authors have addressed all my previous concerns.